# Recent Advances in Dietary Sources, Health Benefits, Emerging Encapsulation Methods, Food Fortification, and New Sensor-Based Monitoring of Vitamin B_12_: A Critical Review

**DOI:** 10.3390/molecules28227469

**Published:** 2023-11-07

**Authors:** Seyed Mohammad Taghi Gharibzahedi, Maryam Moghadam, Jonas Amft, Aysu Tolun, Gauri Hasabnis, Zeynep Altintas

**Affiliations:** 1Institute of Materials Science, Faculty of Engineering, Kiel University, 24143 Kiel, Germany; ato@tf.uni-kiel.de (A.T.); gha@tf.uni-kiel.de (G.H.); 2Institute of Human Nutrition and Food Science, Division of Food Technology, Kiel University, 24118 Kiel, Germany; mmoghadam@foodtech.uni-kiel.de (M.M.); jamft@foodtech.uni-kiel.de (J.A.); 3Kiel Nano, Surface and Interface Science—KiNSIS, Kiel University, 24118 Kiel, Germany

**Keywords:** cobalamin, dietary supplementation, vitamin B_12_ nanoparticles, nanoencapsulation, controlled release, fortified foods, nanobiosensor, biosensing technologies, drug delivery

## Abstract

In this overview, the latest achievements in dietary origins, absorption mechanism, bioavailability assay, health advantages, cutting-edge encapsulation techniques, fortification approaches, and innovative highly sensitive sensor-based detection methods of vitamin B_12_ (VB_12_) were addressed. The cobalt-centered vitamin B is mainly found in animal products, posing challenges for strict vegetarians and vegans. Its bioavailability is highly influenced by intrinsic factor, absorption in the ileum, and liver reabsorption. VB_12_ mainly contributes to blood cell synthesis, cognitive function, and cardiovascular health, and potentially reduces anemia and optic neuropathy. Microencapsulation techniques improve the stability and controlled release of VB_12_. Co-microencapsulation of VB_12_ with other vitamins and bioactive compounds enhances bioavailability and controlled release, providing versatile initiatives for improving bio-functionality. Nanotechnology, including nanovesicles, nanoemulsions, and nanoparticles can enhance the delivery, stability, and bioavailability of VB_12_ in diverse applications, ranging from antimicrobial agents to skincare and oral insulin delivery. Staple food fortification with encapsulated and free VB_12_ emerges as a prominent strategy to combat deficiency and promote nutritional value. Biosensing technologies, such as electrochemical and optical biosensors, offer rapid, portable, and sensitive VB_12_ assessment. Carbon dot-based fluorescent nanosensors, nanocluster-based fluorescent probes, and electrochemical sensors show promise for precise detection, especially in pharmaceutical and biomedical applications.

## 1. Introduction

Vitamins are organic compounds that the body requires in small amounts to maintain normal physiological functions and promote overall health. These micronutrients play essential roles in various biochemical processes, supporting growth, development, metabolism, and overall well-being. Vitamins are classified into two main groups: fat-soluble (A, D, E, and K; Table 1) and water-soluble (C and B-group; Table 2) vitamins. Fat-soluble vitamins stored in the body’s fatty tissues are essential for various functions such as maintaining healthy vision (vitamin A), supporting bone health and immune system function (vitamin D), acting as antioxidants (vitamin E), and aiding in blood clotting (vitamin K) (Table 1) [1,2,3,4,5,6]. Water-soluble vitamins are not extensively stored in the body and should be regularly consumed via balanced nutrition or as a supplement. They play roles in processes such as energy metabolism (B-vitamins), collagen formation and antioxidant protection (vitamin C), and DNA synthesis (vitamins B_9_ and B_12_) (Table 2) [7,8,9,10,11,12,13,14].

A summary of food sources, physiological functions, and RDI values of the B-group vitamins comprising B_1_, B_2_, B_3_, B_5_, B_6_, B_7_, B_9_, and B_12_ is given in Table 2. B-vitamins (recommended daily intake (RDI)) include B_1_ (thiamine; 1.1–1.2 mg), B_2_ (riboflavin; 1.1–1.3 mg), B_3_ (niacin; 14–16 mg niacin equivalents), B_5_ (pantothenic acid; 5.0 mg), B_6_ (pyridoxine; 1.3–1.7 mg), B_7_ (biotin; 30 μg), B_9_ (folate; 400 μg dietary folate equivalents), and B_12_ (cobalamin; 2.4 μg) (Table 2). Due to their unique physiological needs and critical roles in growth processes, B-vitamin requirements are higher for vulnerable subjects, such as children (i.e., growth and development, energy metabolism, and nervous system development) and pregnant women (i.e., fetal development, cell division, blood production, energy and nutrient transfer, maternal health, and breastfeeding) (Table 2).

Because of the absence of vitamin B_12_ (VB_12_) in plant-based food sources, vegans face a significant challenge in obtaining an adequate supply of this vitamin. To address this deficiency, they often rely on consuming plant-based foods that have been fortified with VB_12_. However, factors such as socioeconomic status (including income and education), ethical considerations, cultural practices, and religious beliefs can further limit people’s access to VB_12_ from animal-derived products. This complex interplay of dietary choices and external influences underscores the need for careful nutritional planning and consideration among those following a vegan lifestyle [15]. Furthermore, aging, malabsorption disorders (such as Crohn’s disease and pernicious anemia), chronic diseases (like diabetes), and some medications (like acid reflux treatments) negatively affect the absorption of VB_12_ [16,17]. A ~62% prevalence of VB_12_ deficiency among vegetarians has been reported. As for pregnant women, children, adolescents, and the elderly, the reported percentages of deficiency varied: 25% to 86%, 21% to 41%, and 11% to 90%, respectively [18].

There is a necessity to extract this water-soluble vitamin from natural resources and encapsulate it into biopolymers as dietary supplements. Extraction techniques for VB_12_ provide numerous advantages, including enhancing its bioavailability, isolating pure VB_12_ from complex matrices, enabling controlled processing to optimize yield, efficient utilization of source materials, supporting industrial-scale production, enhancing stability for extended shelf life, and addressing dietary needs for various populations [19,20]. Additionally, micro- and nanoencapsulation of VB_12_ provide a multitude of benefits, including increased physicochemical stability, enhanced bioavailability, controlled release rate for targeted delivery, taste and odor masking to minimize sensory impact, and diverse possibilities for developing innovative enriched products [21,22]. In recent times, a noticeable trend has emerged towards employing diverse methods for extraction (including fermentation, solvent extraction, enzymatic extraction, super/subcritical fluid extraction, ultrasound- and microwave-assisted extraction, solid-phase microextraction, etc.) and encapsulation (such as spray drying, spray chilling and spray cooling, emulsion technique, fluidized-bed coating, liposome entrapment, coacervation, etc.) of B-group vitamins such as cobalamin [23,24,25]. Developing effective methods for extracting and encapsulating VB_12_ is crucial to ensure its stability and bioavailability, thus enabling the reliable supplementation of this essential nutrient for individuals with dietary restrictions or deficiencies.

A multitude of VB_12_ supplements is readily accessible in the market, each distinguished by the form of VB_12_ they contain, with cyanocobalamin being the predominant form due to its stability, although methylcobalamin and hydroxocobalamin options also exist. These supplements vary in dosage, ranging from a few micrograms to several milligrams, and may include other micronutrients. They are available either as standalone VB_12_ or combined with other B-complex vitamins and multivitamin/mineral supplements. These supplements primarily serve a preventive function, aiming to avert VB_12_ deficiency [26]. Conversely, high-dose VB_12_ treatments are typically reserved for individuals with medically diagnosed deficiency conditions. It is essential to acknowledge that distinct regulatory processes govern medicines and dietary supplements. Medicines undergo a rigorous marketing authorization procedure to ensure their quality, safety, and efficacy, while dietary supplements are categorized as concentrated sources of nutrients or other physiologically active substances [27]. Effective treatment of VB_12_ deficiency can be achieved through various administration routes, including intramuscular (I.M.), oral (including sublingual), or intranasal applications. I.M. injections are often associated with pain, require the expertise of a healthcare specialist for administration, and carry a risk of infection. On the other hand, nasal administration can lead to irritation of the nasal mucosa and may be disrupted by respiratory disorders. However, I.M. administration, regardless of the underlying cause of deficiency, remains the most commonly utilized approach. Recently, the buccal route has offered an effective means of administering VB_12_ tablets, optimizing the bioavailability of the vitamin and ensuring a controlled release for better patient outcomes [28,29,30]. Ali et al. found that carbopol 971p (CP971p)-based buccoadhesive tablets of VB_12_ exhibited superior in vitro drug release, mechanical properties, and mucoadhesive qualities. Additionally, selected hydroxypropyl methyl cellulose (HPMC)/CP971p formulations displayed significantly higher bioavailability, up to 2.7 times that of the Neurotone^®^ I.M. injection [31].

Biosensors offer a compelling approach to detecting VB_12_ due to their high specificity, sensitivity, and real-time monitoring capabilities [32]. Unlike traditional methods, such as chromatography or immunoassays, biosensors utilize biological components like enzymes or antibodies to directly interact with the target molecule, resulting in rapid and accurate detection [33]. This direct interaction enhances selectivity and reduces the need for complex sample preparation, making biosensors particularly advantageous for on-site testing, point-of-care diagnostics, and continuous monitoring [32,33,34]. Additionally, biosensors can be miniaturized and integrated into portable devices, enabling convenient and cost-effective VB_12_ detection in various applications, including healthcare, food safety, and environmental monitoring [34].

To the best of our knowledge, there are no comprehensive overviews on the extraction, encapsulation, fortification, and biosensing techniques of VB_12_. Consequently, the primary aims of this comprehensive review are to bridge this knowledge gap by providing a thorough examination of the various extraction techniques utilized for obtaining VB_12_ from natural sources, evaluating the efficacy of different encapsulation methods to enhance its stability and bioavailability, exploring innovative approaches for food fortification with VB_12_, and delving into the latest advancements in sensor-based monitoring techniques for real-time detection of this essential nutrient. By synthesizing current research across these multidisciplinary domains, this review strives to offer valuable insights into the potential synergies between extraction, encapsulation, fortification, and biosensing strategies, ultimately contributing to a deeper understanding of how to effectively harness the benefits of VB_12_ for both individual health and broader nutritional enhancement.

## 2. Chemical Structure and Dietary Sources of Vitamin B_12_

Breaking down the structure of VB_12_ unveils a tripartite arrangement (Figure 1). At its core, there exists a modified tetrapyrrole engaging a central cobalt ion. This tetrapyrrole-derived ring possesses a peculiar trait—it has undergone a ring-contraction process. As a result, one of the bridging carbon atoms that traditionally connects the four pyrrole rings has been expunged. This transformation yields a contracted, asymmetrical macrocycle, distinct from the tetrapyrrole frameworks recognized in heme and chlorophylls. This condensed ring structure is referred to as a corrin. Moreover, the compound encompasses a nucleotide loop housing an unconventional base—termed dimethylbenzimidazole (DMB)—in the context of VB_12_. Fused to one of the corrin ring’s propionate side chains through an aminopropanol linker, the nucleotide loop extends beneath the plane of the corrin ring. This strategic configuration enables the DMB base to serve as a subordinate ligand for the cobalt ion. Lastly, the trifecta concludes with the third component: the upper ligand binding to the cobalt ion [35,36]. Within VB_12_, this upper ligand can take different biochemical forms including cyanide, adenosyl, methyl, and hydroxyl. Accordingly, cyanocobalamin, adenosylcobalamin, methylcobalamin, and hydroxycobalamin represent the primary cobalamin compounds of VB_12_ [37]. The inclusion of the “cyano” group in VB_12_ arises from the manner in which the molecule is isolated, involving the introduction of cyanide as an aid for its extraction and purification. This process ultimately yields cyanocobalamin as the dominant variant of cobalamin [35].

VB_12_ is notably abundant in animal tissues, rendering it exclusive to animal-derived sources. The primary dietary reservoirs encompass an array of options such as meat, milk, and their products (liver, 26–58 μg/100 g; beef and lamb, 1–3 μg/100 g; eggs, 1–2.5 μg/100 g; dairy products, 0.3–2.4 μg/100 g; and chicken, Trace-1 μg/100 g), various types of fish (salmon, sardine, trout, tuna, etc., 3.0 to 8.9 μg/100 g) and shellfish (10 μg/100 g), and fortified ready-to-eat cereals [38,39]. While VB_12_ is primarily found in animal-derived foods, some plant-based sources have reported trace amounts of VB_12_ analogs. Some reported plant-based sources include certain microalgae species (such as *Spirulina* (*Arthrospira platensis* and *A. maxima*), and *Chlorella* (*Chlorella pyrenoidosa* and *C. vulgaris*)), the red algae of nori or *Porphyra* spp. (such as *Porphyra tenera* and *P. yezoensis*), mushroom (shiitake (*Lentinula edodes*), maitake (*Grifola frondosa*), black trumpet (*Craterellus cornucopioides*), and golden chanterelle (*Cantharellus cibarius*)), soy-based fermented foods (like tempeh and miso), and soil-contaminated vegetables [40,41,42,43]. However, it is important to note that these plant-derived compounds do not provide a reliable source of active VB_12_ for humans. Due to these plant-based compounds’ variability and low bioavailability, individuals following a strict vegetarian or vegan diet are recommended to obtain VB_12_ from fortified foods or supplements to ensure adequate intake [39,40]. Furthermore, some bacteria species were used for the industrial production of VB_12_ in fermentation bioreactors such as *Pseudomonas denitrificans*, *Propionibacterium shermanii*, and *Sinorhizobium meliloti* [44].

## 3. Nutritional Effects and Health Functions of Vitamin B_12_

### 3.1. Absorption Mechanism and Bioavailability Assay

VB_12_ assimilation within the body hinges on the presence of intrinsic factor, a compound secreted by parietal cells located in the stomach. Ingested VB_12_, which is bound to proteins, is released from food under the influence of hydrochloric acid and proteases. Following this release, a complex form between intrinsic factor and VB_12_, facilitating its absorption within the ileum of the small intestine. The efficiency of this absorption process is closely tied to the intrinsic factor’s capacity. Upon being absorbed within the ileum, VB_12_ is transported by the plasma carrier, transcobalamin II, which shuttles it into cells [45,46]. Within these cells, VB_12_ undergoes degradation through lysosomal activity. Subsequently, the liberated VB_12_ molecules traverse into the cytoplasm, where they carry out their essential functions. The journey of VB_12_ within the body culminates in its excretion, primarily through bile. Interestingly, this excreted VB_12_ is reabsorbed and retained by the liver, completing the intricate cycle of VB_12_ dynamics within the body. An excess of VB_12_ is eliminated from the body through urine [46]. Gastric mucosa resection or terminal ileum disease can result in VB_12_ deficiency due to impaired absorption. Pernicious anemia, an autoimmune condition, involves atrophy of the gastric mucosa in the body and fundus of the stomach. This leads to a decrease in parietal cells responsible for producing intrinsic factor, which is essential for VB_12_ absorption [47]. Insufficient levels of VB_12_ occur due to decreased dietary intake and compromised bioavailability caused by factors like pernicious anemia, congenital protein-related defects in VB_12_ absorption, bacterial overgrowth, inflammatory bowel disease, gastrointestinal surgeries, following a vegetarian diet, and some medications that impede VB_12_ absorption [15,48].

In the past, researchers quantitatively assessed VB_12_ absorption rates using radioactive labeling of the corrinoid ring’s cobalt molecule of this vitamin, with cobalt isotopes like ^57^Co, ^58^Co, or ^60^Co. Two main methods were employed: one involved assessing absorption by calculating the unrecovered isotope in feces after administering a labeled VB_12_ dose, while the other measured whole-body retention of an oral dose once the unabsorbed isotope had been excreted in feces. Both methods offered a quantitative estimation of VB_12_ absorption, expressed as the percentage of the administered dose absorbed [49]. The Schilling test served as the standard clinical approach for identifying vitamin malabsorption. It involved an oral dose of ^57^Co-labeled VB_12_ followed by a large intramuscular dose of unlabeled VB_12_ to saturate tissues and ensure measurable isotope appearance in urine within the first 24 h. Less than 10% appearing in urine indicated malabsorption [18,50]. However, this test did not always account for the fact that many elderly individuals can absorb free vitamin but not that bound in food, leading to misdiagnosis. The Schilling test’s limitations, such as radioactivity, urine collection, cost, and diagnostic inaccuracies, led to its decline in clinical practice [50]. A newer qualitative test called the C-CobaSorb test has emerged. This approach involves ingesting three distinct doses of non-labeled VB_12_ at 6 h intervals orally. Subsequently, the alteration in the quantity of VB_12_ attached to the serum transport protein transcobalamin (holotranscobalamin) is gauged [51]. A minimal or lacking rise in holotranscobalamin signifies the potential presence of malabsorption. The second option to the Schilling test involves utilizing carbon-14-labeled VB_12_ (^14^C–VB_12_). This technique entails introducing a ^14^C-labeled substrate into a genetically modified strain of *Salmonella enterica*, resulting in the synthesis of VB_12_ containing a sole ^14^C label situated specifically within the DMB segment of the molecule. After orally ingesting ^14^C–VB_12_, the degree of carbon-14 enrichment in blood, urine, and stool samples can be gauged using accelerator mass spectrometry [52]. Assessing serum ^14^C–VB_12_ levels has been applied to evaluate the absorption and availability of VB_12_ from naturally enriched chicken eggs [53] and fortified bread [54]. However, it is worth recalling that none of these new procedures (i.e., CobaSorb test and ^14^C–VB_12_ utilization) has been comprehensively validated to diagnose VB_12_ malabsorption [50]. Also, Devi et al. have recently synthesized a VB_12_ variant labeled with a stable isotope, [^13^C]-cyanocobalamin, using *S. enterica* by supplying [^13^C_2_]-ethanolamine as the singular carbon source. After undergoing purification and characterization via mass spectrometry, the oral bioavailability of this labeled compound at both higher and lower oral dosages was studied in a state of fasting. Undergoing complete decyanation, [^13^C]-cyanocobalamin transformed into [^13^C]-methylcobalamin. This shift shed light on its metabolic utilization, with the plasma demonstrating distinct early and late absorption phases. Assessing VB_12_ bioavailability involved administering low (2.3 μg) and high (18.3 μg) dosages, resulting in reported averages of 46.2% and 7.6%, respectively. Notably, replenishing VB_12_ stores through parenteral methods before measurement led to a considerable 1.9-fold increase in its bioavailability [55].

### 3.2. Health Benefits and Disease Prevention

VB_12_ plays a vital role in the synthesis of blood cells through two main pathways of DNA synthesis and the metabolism of folate [56]. This vitamin is a necessary cofactor for the key enzyme of methionine synthase, which catalyzes the conversion of homocysteine (Hcy) to an amino acid, methionine. This essential sulfur-containing amino acid is involved in many cellular processes like DNA synthesis. This reaction is essential to generate S-adenosylmethionine (SAM), which is a universal methyl donor used in various methylation reactions in the body, including DNA methylation. Moreover, VB_12_ is involved in another enzymatic reaction through the enzyme methylmalonyl-CoA mutase. This enzyme converts methylmalonyl-CoA, a compound generated from certain amino acids and fatty acids, into succinyl-CoA, an intermediate in the citric acid or Krebs cycle. This reaction is important for energy production and maintaining a functional citric acid cycle. The significance of the methylmalonyl-CoA mutase reaction in red blood cell synthesis lies in the fact that it contributes to preventing the accumulation of toxic levels of methylmalonic acid. High levels of methylmalonic acid can interfere with DNA replication and cell division, particularly in rapidly dividing cells like red blood cell precursors. The inability to properly metabolize methylmalonic acid can contribute to the development of megaloblastic anemia, characterized by large, immature red blood cells [56,57,58]. On the other hand, VB_12_ is also crucial for the appropriate metabolism of folate (vitamin B_9_), which is required to synthesize DNA and its precursors. Folate contributes to one-carbon metabolism, which provides the necessary building blocks for DNA synthesis, cell division, and the production of new cells. VB_12_ aids in activating folate by converting 5-methyltetrahydrofolate (5-methyl-THF), an inactive form of folate, into its active form, tetrahydrofolate (THF). Active THF is then used to provide one-carbon units for various reactions, including the synthesis of thymidine, a DNA building block. The interconnected nature of these processes underscores the vital role that VB_12_ plays in maintaining a functional hematopoietic system and overall blood cell production [59,60,61]. As well, elevated Hcy levels have been associated with an increased risk of cardiovascular diseases (CVDs, including heart disease, stroke, and peripheral vascular disease) due to damage to blood vessel walls, increased blood clot formation, and inflammation. Since VB_12_, along with other B-group vitamins (like B_9_ and B_6_), is involved in maintaining Hcy levels in the body, an adequate level of cobalamin prevents CVDs through the proper function of Hcy–methionine conversion [62,63].

VB_12_ is involved in several mechanisms to maintain cognitive health through the following functions: Hcy regulation, neurotransmitter synthesis, myelin maintenance, antioxidant protection, and nervous system health [64]. Apart from the increased risk of CVDs, cognitive decline and neurodegenerative disorders (like Alzheimer’s disease) can occur at uncontrolled levels of Hcy [65]. VB_12_ is essential for the synthesis of neurotransmitters, which are chemical messengers that transmit signals between nerve cells in the brain. Specifically, this vitamin has a pivotal role in the synthesis of neurotransmitters such as serotonin and dopamine, contributing to mood regulation, memory, and overall cognitive function [66]. VB_12_ is crucial for the maintenance and synthesis of myelin, which is the protective sheath in surrounding nerve fibers, and facilitates efficient nerve signal transmission. A deficiency in VB_12_ can lead to damage to the myelin sheath and impair the transmission of nerve signals, potentially affecting cognitive functions. The body’s antioxidant defense system in the presence of VB_12_ can be promoted due to glutathione regeneration. This antioxidant molecule helps the protection of brain cells from oxidative stress and damage, which are implicated in cognitive decline and neurodegenerative disorders. VB_12_ also supports the health of the nervous system, including the brain and spinal cord [67]. Accordingly, adequate levels of cobalamin in the body are necessary to maintain nerve cells’ structural integrity and function. Visual symptoms are not commonly associated with VB_12_ deficiency. The deterioration of the optic nerve in cases of VB_12_ deficiency is believed to occur primarily through degenerative processes. Nevertheless, instances of optic neuritis, albeit rare, have been documented as a potential consequence of VB_12_ deficiency [68]. Ata et al. highlighted that the deficiency of VB_12_ can lead to optic neuropathy as an initial manifestation. They emphasized that the level of this vitamin in patients with visual disturbance should be checked with a fundus exam. Furthermore, particular attention was directed towards the screening process for individuals who manifest visual disturbances, with a notable focus on the vegan population [69]. Some researchers have assessed the association between insufficient VB_12_ levels and neuropathic ocular discomfort and manifestations among individuals with dry eye disease (DED). Their findings indicate the potential for a positive correlation between VB_12_ deficiency and these ocular symptoms. Additionally, they propose a strategy involving the assessment of serum VB_12_ levels in cases of severe DED, particularly in those exhibiting persistent ocular pain that remains unresponsive to topical treatments [70]. Also, results of a pilot study showed the potential stabilization or reduced rate of functional impairment, neuroretinal degeneration, and microvascular damage in patients with type 1 diabetes with mild signs of diabetic retinopathy who were subjected to a therapeutic regimen involving the administration of citicoline and VB_12_ eye drops over a span of three years [71].

Some theories suggest that VB_12_ deficiency could potentially impact bone quality by promoting bone resorption. The conjunction of VB_12_ with other nutrients (such as vitamin D and calcium) can contribute to maintaining bone health through calcium absorption and bone mineralization. VB_12_ is involved in the synthesis of collagen, providing the framework for bone mineralization, and contributing to bone strength and integrity [72]. However, Zhao et al. concluded that higher VB_12_ concentrations had no significantly beneficial effect on bone health in patients with postmenopausal osteoporosis. Interestingly, subgroup analysis based on ethnicity showed that the correlation between Hcy, VB_12_, and folate with bone health was significant only among the European population. This emphasizes the importance of considering ethnic variations when drawing definitive conclusions about these associations [73].

He et al. reviewed the correlation between serum levels of VB_12_ and vitamin B_9_/VB_12_ B_12_ in pregnant women and the risk rate of gestational diabetes mellitus (GDM). They found that there was a significant association between VB_12_ deficiency and increased GDM risk, showing a prerequisite was to have a balance in the serum levels of vitamins B_9_ and B_12_ [74]. It was explained that since VB_12_ acts as a scavenger of reactive oxygen species (ROS; mainly O_2_^•^−), its deficiency can increase the GDM risk [75]. The use of the oral hypoglycemic drug metformin to treat GDM has increasingly become prevalent. Aroda et al. [76] showed that metformin could curtail the absorption of VB_12_. Consequently, women with GDM who are treated with metformin may exhibit diminished levels of VB_12_ [75]. However, it is important to emphasize that metformin’s impact seems to be specific to the absorption of VB_12_, without affecting its active form known as holotranscobalamin. This finding suggests that a reduction in the absorption of VB_12_ may not necessarily result in alterations to the levels of its bioactive form [74].

Given the absence of conclusive evidence establishing a direct causal link between VB_12_ and cancer, addressing low VB_12_ levels in individuals with cancer becomes crucial. This proactive approach aims to mitigate the risk of neurological impairment and enhance the capacity to endure cancer treatments, including chemotherapy. Nevertheless, it is important to note that an excessive intake of VB_12_ could potentially elevate the risk of cancer [77]. Luu et al. conducted a study that highlighted a dosage-dependent correlation between dietary VB_12_ consumption and an increased likelihood of lung cancer advancement. This association was particularly prominent among male participants, especially those afflicted with adenocarcinoma, and individuals undergoing a follow-up period of two years or less [78]. This revelation, combined with the findings derived from comprehensive cohort studies investigating VB_12_ supplementation [79,80], reinforces the notion that the inclusion of high doses or elevated levels of VB_12_ may not yield benefits in terms of lung cancer prevention strategies. In fact, this approach could potentially result in adverse effects, particularly among male individuals [78].

## 4. Emerging Encapsulation Methods of Vitamin B_12_

The encapsulation of VB_12_ holds significant scientific merit, primarily stemming from its susceptibility to environmental degradation and the intricate physiological dynamics underlying its absorption. VB_12_ is notably labile, subject to degradation in the presence of factors like light, heat, oxygen, and moisture. Encapsulation provides a protective microenvironment that shields the vitamin from these detrimental influences, thereby preserving its structural integrity and bioactivity [52,81,82]. Furthermore, the controlled release properties conferred by encapsulation are of paramount importance. By governing the release kinetics, encapsulation can synchronize the availability of VB_12_ with the physiological processes responsible for its absorption, facilitating optimized uptake and utilization. This becomes particularly relevant in scenarios like food fortification and nutraceutical formulations, where precise dosing and prolonged bioavailability are critical for ensuring therapeutic or nutritive efficacy [83,84,85,86]. Therefore, encapsulation strategies not only safeguard VB_12_ from degradation but also harness its delivery dynamics to align with its intricate metabolic pathways, substantiating its significance in diverse applications. Encapsulation techniques for VB_12_ encompass a diverse array of approaches, spanning both micro- and nanoencapsulation strategies (Table 3). These methodologies capitalize on distinct mechanisms to confer protection, controlled release, and enhanced bioavailability of VB_12_.

### 4.1. Microencapsulation

#### 4.1.1. Single-Core Microcapsules

Carlan et al. compared the ability of several wall materials (i.e., sodium alginate, carrageenan, gum arabic, maltodextrin, modified starch, xanthan, and pectin) to microencapsulate VB_12_ (3.17–6.67 μm) by the spray-drying process through the controlled release assessment of samples both fresh and after 4 months storage. All microparticles were spherical in shape with smooth (with sodium alginate, carrageen, maltodextrin, and pectin) and rough (with others) surfaces. Although the powder yield was not significant (27–50%), they represented good storage stability properties (at least for 120 days) with the possibility of releasing VB_12_ over various time periods [87]. In another study, these researchers prepared modified chitosan-based microparticles (3–8 μm) containing VB_12_ (1–5% *w*/*w*) using spray-drying and evaluated the controlled release of it under the in vitro simulated gastric conditions. A ~57% powder yield was obtained for microparticles with a regular round shape and a smooth surface. The release data of VB_12_ at 37 °C from the surface of these particles could well fit with the Weibull kinetic model. The storage stability analysis showed a superior retention of VB_12_ after 3 (<10%) and 6 (<20%) months [88]. Estevinho et al. applied a cyanobacterial extracellular carbohydrate polymer alone or combined with gum arabic to microencapsulate VB_12_ using a spray-drying process. A higher product yield along with a slower release rate was obtained when spherical microparticles (<~8 μm) with very rough surfaces were produced with a conjugation of gum arabic and the cyanobacterial polymer (1:1). The release kinetic of VB_12_ from superior microparticles showed that the Weibull model had the best fit to the experimental data [89].

Coelho et al. designed zein-based microstructures loaded with VB_12_ using electrospinning and spray-drying methods. Electrospinning produced microstructures in terms of films, 3 μm microbeads, and electrospun fibers. The encapsulation efficiency (EE) of the corresponding microstructures were 90, 91, and 100%, respectively. However, spray-drying produced microparticles with wrinkled surfaces with a favorable product yield (67–83%) and EE (71–95%). They also found that the Weibull kinetic model was the best fitting with the actual release values of VB_12_ [90]. Figure 2 illustrates microscopic images of 10–20% zein-based nanostructures containing VB_12_. The optimal outcome for microstructures with 10% *w*/*w* zein is shown in Figure 2C, exhibiting a homogeneous matrix with minimal pore formation. Figure 2D shows nanostructures composed of 20% zein and 5% VB_12_, using a flow rate of 0.2 mL/h at a distance of 7 cm. Furthermore, an increase in the distance to 10 cm resulted in the formation of electrospun fibers accompanied by some shrunken microbeads (Figure 2E). Figure 2F reveals that increasing the distance to 15 cm led to the development of a matrix with a reduced number of pores. The microscopy images in Figure 3 reveal microparticles obtained through the spray-drying process, displaying spherical-like shapes with wrinkled surfaces. Some minor agglomeration of the microstructures is noticeable. Interestingly, the results indicated that particle size remains consistent across the various samples, regardless of the amount of loaded VB_12_. This suggested a common microencapsulation mechanism responsible for microparticle production [90].

Recently, the effectiveness of liposomes loaded with methyl jasmonate mixed with VB_12_ has been demonstrated on the prevention of chilling injury (CI; translucency and browning in tissue adjacent to the core) in ‘Queen’ pineapples. This delivery system increased the ascorbic acid content and antioxidant activities of tissue adjacent to the core of refrigerated pineapples [91]. Sugandhi et al. have recently developed spherical microparticles (0.59 μm) of VB_12_ based on the mixture of bovine serum albumin (BSA as an encapsulant) and pullulan (as a mucoadhesive agent) for pulmonary drug delivery using spray-drying. A good in vitro bioavailability (64.1%) of VB_12_ from microparticles was recorded while pharmacokinetic analyses on male Wister rats via an intratracheal administration showed the increased permeability and in vivo bioavailability (4.5 folds) of this vitamin [92]. A gelatin-based microsphere containing VB_12_ and decorated with a carbon dot (CD) metal–organic framework (MOF) was designed for concurrent usage in pH sensing and wound closure. This system showed the powerful antimicrobial potential (against *Escherichia coli* and *Staphylococcus aureus*) and the proliferation of fibroblast L929 cells for tissue regeneration in a time-dependent pattern [93].

Mazzocato et al. realized that the stability of this vitamin was significantly improved after its loading into solid lipid microparticles (SLMs; with 0.1 and 1% VB_12_ and 0, 2.5, and 5% soy lecithin) using the spray-chilling technique. This approach is low-cost and does not need high temperatures or organic solvents to encapsulate bioactive ingredients such as VB_12_. The shape, average size, product yield, EE, and storage stability of SLMs were spherical with smooth surfaces, 13.28–26.99 μm, 80.7–99.7%, 76.7–101.1%, and >91.1% at 4 months, respectively. They proved that a combination of soy lecithin and the spray-chilling technique could improve the controlled release and provide better stability of VB_12_ [94]. Akbari et al. optimized the effect of pectin and whey protein concentrate (WPC) levels and pH value on the physicochemical characteristics of pectin–WPC complex carriers containing VB_12_. A combination of 1:6.47 pectin:WPC and pH = 6.6 led to the highest EE (80.71%), stability (85.38%), viscosity (39.58 mPa·s), solubility (65.86%), and the lowest particle size (7.07 μm). Fourier transform infrared (FTIR) spectra confirmed the presence of amide bonds between pectin and WPC to form the strong complex coacervate for microencapsulating VB_12_ [24].

Two conventional and membrane emulsification techniques were also employed to fabricate water-in-oil-in-water double (W/O/W) emulsions encapsulating either *trans*-resveratrol or vitamin B_12_ using stabilizers of sodium carboxymethyl-cellulose and Tween 20. The membrane emulsification formed more uniform droplets and better EE of core materials compared to the conventional one using mechanical agitation [95]. Nollet et al. also investigated the emulsification process and rheological behavior on the EE of vitamin B_12_ in double W/O/W emulsions using two stabilizers of sodium caseinate and gum arabic. They realized that a laminar flow even at very high shear stress resulted in a ~100% encapsulation rate, indicating a correlation between the fluxes of water and the EE. Gum arabic-stabilized double emulsions compared to sodium caseinate-stabilized ones were less sensitive to an osmotic unbalance and had better resistance to water transfer. They explained that these double emulsions can efficiently encapsulate hydrophilic bioactive compounds with a molar weight of ≥20 kDa [96]. Another study assessed the effect of soybean lipophilic protein–methyl cellulose (SLP:MC) complex on W/O/W emulsions loaded with VB_12_. Such a complex improved the viscoelasticity of W/O/W emulsions by developing a strong gel network on the surface of droplets, providing a high EE during storage. Moreover, a ratio of W_1_/O:W_2_ of 4:6 accompanied by an SLP:MC of 3:1 could guarantee steady behavior in releasing VB_12_ under in vitro digestion [97]. A gelatin-g-poly (acrylic acid-co-acrylamide)–montmorillonite superabsorbent hydrogel was also fabricated to improve the in vitro release of VB_12_. Results showed that the release rate of VB_12_ after 6 h in artificial gastric and intestinal fluids was 42% and 80%, respectively [98].

#### 4.1.2. Double- and Multiple-Core Microcapsules

Bajaj et al. using spray-drying to co-microencapsulate vitamins B_12_ and D_3_ into an optimal biopolymeric matrix comprising gum acacia, Hi-Cap^®^ 100, and maltodextrin with a ratio of 38:60:2. The spherical micro-particles produced with a smooth surface could well maintain vitamins’ stability with a slow release rate under the in vitro digestion conditions. A remarkably enhanced bioavailability of VB_12_ (151%) and vitamin D_3_ (109%) compared to the control was reported. Interestingly, the in vivo studies in rats revealed a slower release rate and more superior absorption potential of encapsulated VB_12_ than vitamin D_3_ in serum [99]. Estevinho et al. co-encapsulated VB_12_ and vitamin C with the different encapsulating agents (i.e., chitosan, modified chitosan, and sodium alginate) using the same technique under the following conditions: at 4 mL/min (15%) solution flow rate, 32 m^3^/h (80%) air flow rate, 6.0 bar air pressure, and 120 °C inlet temperature. They found a ~45% powder yield containing ~3 μm microparticles. Chitosan-based microparticles showed rougher surfaces and a slower release rate of vitamins compared to microparticles developed with other biopolymers. This study demonstrated that the selection of encapsulating agents is necessary for better-controlled release of these vitamins [100]. The co-encapsulation of epigallocatechin-3-gallate (EGCG, 0.5–5% *w*/*w*) and VB_12_ (0.5–5% *w*/*w*) into zein protein (1–30% *w*/*v*) was performed using the electrospinning technique. Based on changing the solution conductivity or increasing the interaction between the biopolymer and the solvent, they realized that the formed particles’ shape was a function of the zein concentration. Results also showed that the maximum EE belonged to microstructures composed of 1% *w*/*w* of bioactive constituents and 30% *w*/*v* of zein. Figure 4 shows a consistent pattern for the release behavior of the samples, except for the EGCG samples containing 1% and 5% *w*/*v* zein, which display rapid initial release. In the case of all other samples, two distinct phases can be discerned: the release phase, marked by a notable rise in core release, and the stabilization phase, characterized by a plateau in the release rate. Moreover, the release kinetic analysis revealed that the Weibull model had the best fit for the storage-dependent release rate [83].

Roy et al. developed new boronic acid-derived amphiphile-based gel emulsions to efficiently encapsulate VB_12_ and anticancer drug doxorubicin [101]. Guchhait et al. also applied tripeptide-based nontoxic hydrogelators to co-encapsulate VB_12_ and doxorubicin. The release rate of these bioactive constituents was highly dependent on the medium pH. The release ability in buffer solutions (pH 7.46) containing VB_12_ was more than doxorubicin. After 48 h, 80–85% of VB_12_ and doxorubicin at pH values 7.46 and 4.5 were released from their respective hydrogels, respectively. The enhanced release of doxorubicin at the acidic pH of 4.5 over the physiological pH of 7.46 suggests that these hydrogels hold potential as carriers for cancer drugs (Figure 5). No remarkable lethality towards normal human lymphocytes was detected up to 25 μg/mL. Moreover, the designed delivery systems presented robust anticancer activity against human breast cancer cell MCF-7 and human lymphocyte cell HLC [102].

A stable double emulsion (~96% in a 30 d storage) containing highly nutritious ingredients (i.e., vitamins B_6_, B_12_, and C, as well as black chokeberry pomace extract in the aqueous phase and vitamins A and D_3_ in the oil phase) was prepared using a two-step mechanical emulsification. An EE (75.0–99.3%) and encapsulation stability (74.0–95.9%) of all the bioactive compounds with controlled release during digestion were recorded. However, only 20% of VB_12_ was released at the end of the gastric digestion stage, while this vitamin, similar to other vitamins, was entirely released at the duodenal phase end. The double emulsion as an efficient delivery system can be used in the formulation of elderly people’s diets to increase the bioavailability of bioactives [103]. Pattnaik et al. evaluated the effect of ultrasonication and wall materials on the encapsulation of multi-vitamins (i.e., A, D, B_9_, and B_12_) in a two-stage emulsification process. Except for vitamin B_9_, other vitamins had a significant entrapment efficiency (>96%) in the developed lipid-based double emulsion template. The ultrasonication time had no considerable effect on the EE of vitamins, although the particle size was reduced by increasing this operating parameter. They also showed that low-fat biscuits fortified with this delivery system provided the highest retention of vitamins compared to the unencapsulated ones [104].

### 4.2. Nanoencapsulation

#### 4.2.1. Nanovesicles

Nanovesicles (NVs) have emerged as highly versatile carriers for a wide range of bioactive compounds and pharmaceuticals. Their distinctive structure, characterized by an aqueous core encased within a lipid layer, makes them exceptionally well-suited for various applications. This structural blend of hydrophilicity and lipophilicity opens the door to a multitude of uses. Also, NVs offer several practical advantages, including exceptional biocompatibility, adaptability in terms of size, and flexibility in composition. These attributes make them an ideal choice for a diverse array of applications. In recent years, NVs have found valuable roles in delivering VB_12_ within industries such as cosmetics, food, pharmaceuticals, and biotechnology. This utilization underscores their growing importance in enhancing the delivery and efficacy of bioactive compounds across different sectors.

Based on the synergic activity of VB_12_ with antibiotics, Marchianò et al. developed three types of lyophilized NVs–niosomes, positively-charged niosomes, and liposomes incorporating this vitamin. To safeguard these structures during freeze-drying, they introduced a protective substance, 20% maltodextrin, which helped shield the NVs from the harsh conditions of the process. Interestingly, the size of the NVs remained largely unaffected after the addition of VB_12_ prior to lyophilization. They observed a significant enhancement in the EE and loading capacity of VB_12_ in the nanostructures that contained PEG400, with a notable 70% improvement and a loading capacity of 100 mg/g. Furthermore, the researchers concluded that the inclusion of glycerol in the NV formulation had a substantial impact on increasing the antimicrobial effectiveness of these delivery vehicles. This finding underscores the potential of NVs, particularly in combination with VB_12_, as effective carriers for antibiotics and antimicrobial applications [105]. Guillot et al. achieved a significant breakthrough by designing innovative lipid vesicles, which typically ranged in size from 141 to 283 nm. These vesicles were engineered to encapsulate VB_12_ effectively. This encapsulation strategy aimed to enhance the vitamin’s ability to penetrate the skin, particularly for the purpose of inhibiting chronic inflammatory skin conditions such as atopic dermatitis and psoriasis [106]. In a separate study, the same researchers formulated VB_12_ within ultra-flexible lipid vesicles, known as transfersomes and ethosomes, with sizes ranging from 140 to 307 nm. These lipid vesicles were designed for targeted delivery to skin cells by freeze-drying in the absence and presence of lactose and sorbitol as cryoprotective agents. According to the visual inspection (Figure 6), the use of sorbitol resulted in a collapsed cake and a suboptimal final product. The collapsing process implies only minor or cosmetic irregularities that do not affect patient safety or product efficacy. Consequently, sorbitol was deemed a suboptimal cryoprotectant. In contrast, lactose produced a uniform product with no signs of damage (Figure 6). What makes this research even more intriguing is their use of a solid microneedle array for skin pre-treatment. This pre-treatment method proved effective in improving the quantity of the drug that could potentially reach the systemic circulation. This novel combination of vesicles and microneedle technology opens up exciting possibilities for enhanced transdermal drug delivery [107].

García-Manrique et al. conducted a study to assess how the composition of the hydrating solution (including Milli-Q water, glycerol, and PEG-400) and the molecular weight of the loaded compounds (such as rhodamine, vitamin C, and VB_12_) impacted both the particle size and EE of niosomes (Figure 7A). The results revealed that the particle size was significantly influenced by the specific composition of the hydrating solution and its interaction with the molecular weight of the bioactive compounds. Notably, an increase in the particle size of niosomes correlated with higher EE and greater loading capacity. Furthermore, the choice of hydrating solution composition had a profound impact on the bilayer packing and physical properties of the niosomes, demonstrating its significant role in shaping these lipid-based vesicles [108]. Earlier, some researchers [109] aimed to produce nanostructures known as Small Unilamellar Vesicles (SUVs; <100 nm), and to load them with various bioactives, including VB_12_, tocopherol, and ergocalciferol. The researchers started with lipidic microstructures referred to as Multilamellar Large Vesicles (MLVs). The study focused on identifying suitable formulations, optimizing sonication protocols, and developing these nanoliposomes (Figure 7B). Remarkably, the research led to the successful creation of SUVs with diameters ranging from 40 to 51 nm, originating from MLVs with diameters spanning between 2.9 and 5.7 μm. These SUVs exhibited high EE for VB_12_ (56%), alpha-tocopherol (76%), and ergocalciferol (57%), indicating their effectiveness in encapsulating these vitamins. Furthermore, stability tests demonstrated that the chosen lipid composition was capable of preserving the integrity of these NVs and their vitamin content for over 10 days, particularly when incubated under simulated extracellular environmental conditions [110].

#### 4.2.2. Nanoemulsions

Nanoemulsification represents a cutting-edge approach aimed at enhancing the solubility of hydrophobic substances in water, facilitating precise delivery within the gastrointestinal system [111]. Nanoemulsions (NEs), characterized by their minuscule droplet size, offer a range of appealing qualities. These include heightened physical stability, exceptional optical clarity, heightened aqueous solubility, and notably improved bioavailability, both in vitro and in vivo, of bioactive compounds [111,112,113,114,115]. As such, NEs hold great promise as a vehicle for more effective delivery of functional ingredients and pharmaceuticals [116,117].

In recent studies, NEs served as carriers or delivery systems for water-soluble VB_12_. Karbalaei-Saleh et al. employed response surface methodology to fine-tune the spontaneous emulsification process for nano-encapsulating VB_12_. Their findings pointed to an optimal formulation comprising 6.5% sunflower oil, 9.6% Tween 80, and 13% VB_12_. This specific blend yielded the highest levels of EE, viscosity, and VB_12_ content, while simultaneously achieving the lowest pH, turbidity, *p*-anisidine index, particle size, and polydispersity index (PDI) [118]. In a similar vein, an optimization technique was applied to enhance VB_12_-loaded double NEs for fortifying skim milk using ultrasound-assisted nanoemulsification. The study yielded impressive results with two optimized formulations of W_1_/O/W_2_ emulsions. In the first formulation, a 12% concentration of the W_1_/O phase, combined with 1 min of sonication, resulted in an EE of 88.85%, droplets sized at 57.14 nm, and a low PDI of 0.19. The second optimized formulation achieved an EE of 79.70% by using a 10% concentration of W_1_/O in W_1_/O/W_2_ emulsion and extending sonication to 2 min (Figure 7C,D). This led to a slightly larger droplet size of 65.29 nm with a PDI of 0.21. Both of these optimized NEs exhibited exceptional stability over one week at 37 °C, with no observable phase separation. However, it is important to note that the droplet size, sedimentation index, and PDI experienced slight increases during storage, resulting in a decrease in EE [119]. Çoban et al. embarked on the development of a new, stable NE featuring alpha-lipoic acid and VB_12_. This NE was designed for the treatment and prevention of conditions like diabetic neuropathy, peripheral neuropathy, and various neurological disorders. Their research demonstrated impressive attributes, including high colloidal stability across different temperatures and pH levels. The researchers achieved outstanding EE values, including 93.80% alpha-lipoic acid and 110.65% VB_12_. These promising results were obtained through the preparation of castor oil-in-water NEs using a magnetic stirring method [120]. Cheng and Compton introduced a fascinating approach involving single VB_12_ nanodroplets to facilitate the reduction of oxygen in a neutral buffer. They, through the utilization of the nano-impacts method, observed electron transfer events to individual VB_12_ nanodroplets. This investigation unveiled the intricate mechanism behind oxygen reduction mediated by single VB_12_ droplets. This mechanism operates through the reduction of CoIII in VB_12_ to both CoII and CoI states, accomplished via one or two electron transfers. Subsequently, this is followed by the four-electron reduction of oxygen. This breakthrough sheds light on a novel and detailed process that could have wide-reaching implications in various fields, particularly in the realm of catalysis and energy conversion [121].

#### 4.2.3. Nanoparticles

Nanoparticles (NPs) exhibit the potential to adhere to the intestinal mucus layer through non-specific interactions and traverse this barrier to engage with the intestinal epithelium. Furthermore, they have been demonstrated to facilitate the translocation of nutraceuticals across the intestinal epithelium through both paracellular and transcellular routes. This capability opens up exciting prospects for improving the absorption and delivery of essential nutrients and bioactive compounds in the field of nutrition and pharmaceuticals [22,122]. Soy protein NPs, created through a cold-gelation method, were harnessed from the encapsulation of VB_12_ with the goal of enhancing its intestinal transport and absorption. It was observed that the intestinal transport of VB_12_ experienced a significant up to 4-fold increase after being encapsulated within soy protein NPs, which had a size of 30 nm. This improvement in transport was attributed to the NPs being taken up by various pathways, including clathrin-mediated endocytosis and micropinocytosis [123]. This innovative approach has the potential to enhance the bioavailability and effectiveness of VB_12_ supplementation, offering promising implications for health and nutrition. Liu et al. [124] engineered protein–lipid composite NPs for the efficient delivery of hydrophilic nutraceuticals, including VB_12_. They enhanced the stability and efficacy of these NPs by succinylating the protein outer layer, increasing surface charge, and extending the succinate chains on the NP surface. This succinate-mediated crosslinking significantly reduced the leakage rate of VB_12_ during 30 days of storage (down to 4.5%) and remarkably improved its uptake efficiency (by more than 20 times). These NPs exhibited an ideal release rate for VB_12_ over 10 h in simulated intestinal fluid. Compared to free VB_12_, these NPs were more effective in compensating for deficiencies in a rat model, marking a promising development in the field of nutraceutical delivery [124]. In a prior study, these researchers employed a similar composite NP approach to enhance the EE of VB_12_ (69%). They also achieved controlled release in simulated gastrointestinal conditions, along with improved uptake and transport efficiency in model Caco-2 cells. When administered orally, these NPs outperformed free VB_12_ supplements, significantly reducing the levels of methylmalonic acid in rats without inducing any toxicity effects. This research highlights the potential of these NPs for enhancing the bioavailability and therapeutic efficacy of VB_12_ [125]. Ramalho et al. took a different approach by utilizing poly(lactic-co-glycolic acid; PLGA) NPs to co-encapsulate vitamins B_9_ and B_12_, aiming to enhance their stability and oral bioavailability. The organic NPs, approximately 190 nm in size, achieved high EEs of 89% for vitamin B_9_ and 71% for VB_12_. These powdered NPs exhibited excellent stability (at least 8 weeks) at room temperature and maintained high in vitro bioaccessibility and bioavailability rates in the gastrointestinal environment, showcasing their potential for improving the delivery of these vitamins [126].

Singh et al. showcased the utility of VB_12_ as a surface functionalization material for solid lipid nanoparticles (SLNs; Figure 7E). This novel approach aimed to enhance the oral bioavailability of the encapsulated drug while ensuring high cellular viability and resilience to the harsh conditions of the gastrointestinal tract. Results revealed that the active mechanism associated with VB_12_ stearate–amphotericin B-SLN synthesized using the double-emulsion solvent evaporation method was energy-dependent and involved clathrin-mediated pathways. Moreover, they made a crucial discovery regarding the non-invasive oral administration of this formulation as a promising and highly efficient strategy for combatting leishmaniasis. This therapeutic approach did not induce cellular toxicity or severe disruption in membrane viscoelasticity. The formulation’s consistent mucus retention ability significantly impacted its systemic absorption and subsequent bioavailability [127]. The other research group utilized sonication to fabricate nano-VB_12_ (120–180 nm) and nano-penicillin (70 nm) with the enhanced bio-functional effects like antimicrobial activity [128]. Also, the reduced size of organic NPs of VB_12_ improved the penetration depth within the skin and the underlying tissue to induce antioxidative effects by scavenging intracellular superoxide anion radicals [128,129]. In a study by Verma et al. [130], they explored the potential of layer-by-layer coated calcium phosphate NPs for oral insulin delivery. This approach involved using VB_12_-grafted chitosan and sodium alginate as cationic and anionic polyelectrolytes, respectively. Remarkably, VB_12_ served as a pH-sensitive and targeting ligand, significantly enhancing insulin’s oral bioavailability.

**Figure 7 molecules-28-07469-f007:**
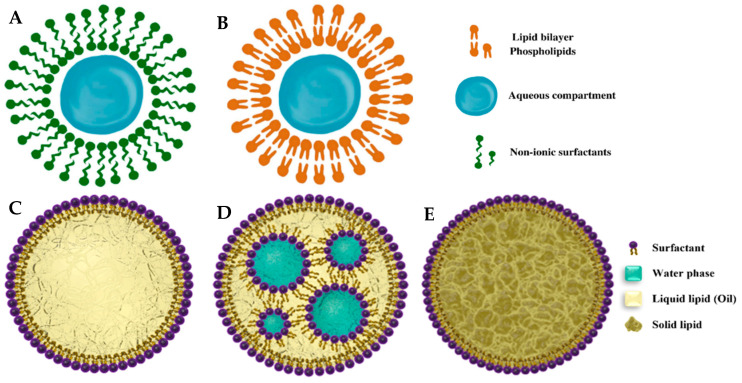
Schematic images of some VB_12_-nanoencapsulating systems including niosome (**A**), liposome (**B**), O/W emulsion (**C**), W/O/W emulsion (**D**), and SLNs (**E**). Reprinted from [131,132].

## 5. Food Fortification

A variety of strategies have been globally adopted to tackle micronutrient deficiencies, including biofortification, food fortification, and supplementation. These approaches are designed to boost the nutritional value of both food and supplements, with the overarching goal of enhancing public health and addressing micronutrient deficiencies. Specifically, the fortification of food products, especially those derived from plant sources, with free and encapsulated VB_12_ can be seen as a cost-effective and natural means to ensure sufficient intake of this essential micronutrient, particularly for populations vulnerable to VB_12_ deficiency. This approach holds promise in promoting better health outcomes and mitigating the risks associated with inadequate VB_12_ intake [17,122,133].

### 5.1. Direct Enrichment of Vitamin B_12_

Modupe and Diosady [134] recently enhanced salt with a combination of iron, iodine, folic acid, and VB_12_. They introduced VB_12_ into the iron premix using four different approaches: co-extrusion with iron, surface spraying onto iron extrudate, inclusion in the color masking agent (TiO_2_), and addition to the outer coating. Among these methods, coextrusion emerged as the most efficient, thanks to its simplicity of production and the stability of the fortificants. In the optimal formulation, over 98% of VB_12_, 93% of folic acid, and 94% of iodine were retained even after a 6-month storage period. Researchers employed a co-crystallization technique with various hydrocolloids to create sugar cubes fortified with VB_12_ [135]. They discovered that the highest retention of VB_12_, reaching 97.5%, was achieved when using 2.5% gum acacia (*w*/*w* of sucrose). The resulting product not only exhibited reduced crystallinity compared to pure sucrose crystals but also displayed a gradual release of encapsulated VB_12_ during in vitro digestion testing. Furthermore, kinetic studies on VB_12_ degradation revealed exceptional stability, with a half-life of 693 days at a temperature of 25 °C and 30% relative humidity (RH) [135]. This approach holds promise for enhancing the shelf life and nutritional value of fortified sugar products. In a separate study by Vora et al. [136], the prospect of mitigating folate and VB_12_ deficiency through vitamin-enriched tea was examined. They concluded that the daily consumption of vitamin-fortified tea for a period of two months had a positive impact on the health of young women of childbearing age in Sangli, India. This intervention resulted in notable increases in their serum folate, serum VB_12_, and hemoglobin concentrations.

Bajaj and Singhal [137] examined the fortification of whole wheat flour with vitamins B_12_ and D_3_. During storage, the degradation of both vitamins in whole wheat flour was notably accelerated with higher storage temperatures. However, the RH primarily influenced the degradation rate of vitamin D_3_. Their findings also revealed the order of VB_12_ retention in bakery products fortified with cyanocobalamin as follows: flat-bread (chapattis) retained the highest at 90.6%, followed by bread at 79.7%, cake at 35.8%, cookies at 24.5%, and oil-fried flat-bread (poori) at 13.9%. This hierarchy provides valuable insights into the suitability of different bakery items as carriers for VB_12_ fortification [137]. Wang et al. undertook a unique approach by initially biosynthesizing VB_12_ (7.9–8.9 μg/100 g fresh weight) and dextran through the fermentation of rice bran or soy flour using *Propionibacterium freudenreichii* DSM 20271 (vitamin producer) and Weissella confusa A16. They subsequently utilized these compounds to fortify bread products. Their findings revealed that the fortified bread displayed improved texture and sensory qualities compared to the control, along with an extended microbial shelf life. Importantly, they highlighted that consuming 100 g of fortified bread would fulfill the RDI for VB_12_, underlining the potential of such products to enhance nutritional outcomes [138]. In a similar study by Xie et al. [139], they utilized fermentation with *P. freudenreichii* DSM 20271 and *Levilactobacillus brevis* ATCC 14869 to biosynthesize VB_12_ (greater than 300 ng/g) from eleven cereal, pseudo-cereal, and legume materials. The highest production of VB_12_ was achieved through the fermentation of rice and buckwheat brans. This research suggests that these fermented materials hold promise for effectively fortifying plant-based foods with VB_12_, addressing the needs of individuals following vegetarian and vegan diets. Hemery et al. [140] conducted a study assessing the impact of controlled storage conditions (25 °C or 40 °C; 65% or 85% RH) and packaging types (paper bags or multilayer aluminum/polyethylene terephthalate (PET) bags) on the microbial load and stability of vitamins B_9_ and B_12_ in fortified wheat flour over a 6-month period. They regularly examined the content of vitamins B_9_ and B_12_, as well as microbial quality. In flours stored in multilayer bags (impermeable to oxygen and humidity), no significant losses of vitamins B_9_ and B_12_ were observed, regardless of temperature and humidity levels. The microbial quality of flours was reduced when they stored in permeable paper bags at 85% RH. The content of vitamins B_12_ (37–51%) and B_9_ (47–78%) in flours packed in permeable paper bags was significantly diminished after 6 months at 65% RH. This underscores the critical importance of packaging choice, depending on environmental conditions, to maintain the stability and quality of fortified wheat flour [140].

### 5.2. Enrichment with Vitamin B_12_ Capsules

Recent studies have explored the fortification potential of dairy products with VB_12_. In one study by Melo et al., the aim was to develop yogurts fortified with microencapsulated forms of VB_12_ (methylcobalamin and cyanocobalamin) to enhance their shelf-life stability. Microcapsules were created by spray-drying a polymeric material derived from maize starch with 1% (*w*/*v*) VB_12_. Full-fat stirred yogurts, containing 3.25% milk fat, were enriched with both free and microencapsulated VB_12_ at a concentration of 50 μg/175 g and stored at 4 °C for eight weeks. The synthetic form of cyanocobalamin demonstrated greater stability as a fortificant throughout its shelf life. Encapsulation techniques proved to be a viable method for increasing the stability of methylcobalamin in fortified yogurts [141]. As stated earlier, Zaghian and Goli [119] fortified skim milk with VB_12_-loaded double NEs prepared using ultrasound-assisted nanoemulsification. Also, water-in-oil-in-water double emulsions were employed as carriers to encapsulate VB_12_, achieving an impressive EE of 96%, which were incorporated into cheese formulations. These innovative delivery systems effectively prevented vitamin loss during in vitro gastric digestion, with less than 5% of VB_12_ being released. Moreover, fortifying cheese with VB_12_ had a remarkable impact. It reduced VB_12_ losses in whey and significantly increased its retention rate in cheese, elevating it from 6.3% to over 90%. This approach demonstrates the potential to enhance the nutritional content and stability of cheese products through VB_12_ fortification [142]. However, Jurado-Guerra et al. examined the combined effect of free VB_12_ and melatonin on whole milk stirred yogurt. The best-performing yogurts were fortified with 3 μg/L of VB_12_ and 5 mg/L of melatonin, maintaining similar rheology and syneresis values to the control yogurt [143]. These findings offer insights into enhancing the nutritional value and stability of dairy products through fortification with encapsulated forms of VB_12_ and other bioactive compounds. Pattnaik et al. undertook the development of low-fat biscuits fortified with vitamins A, D, B_9_, and B_12_, in both free and nano-encapsulated forms. Notably, the cumulative release of all these vitamins in biscuits containing encapsulated NPs was relatively low. This suggests excellent bioavailability and thermal stability of the fortified food product, highlighting the potential for these biscuits to serve as a nutritious and long-lasting dietary option [144]. In another study, Pattnaik et al. also developed low-fat biscuits fortified with liposomes containing multi-vitamins (A, D, B_9_, and B_12_), indicating more retention of vitamins compared to the un-encapsulated ones [104].

## 6. Methods for Extracting Vitamin B_12_ from Capsules

Extracting VB_12_ from micro- and nanocapsules can be achieved through several methods (such as dissolution and filtration, ultrasonication, centrifugation, chemical extraction, enzymatic digestion, and supercritical fluid extraction), depending on the nature of the capsules and the desired outcome. When dealing with micro or nanocapsules engineered for controlled release, the extraction of VB_12_ can be accomplished through a dissolution and filtration method. In this procedure, the capsules are deliberately dissolved in a suitable solvent chosen for compatibility with the VB_12_ and the capsule material. The resulting solution is then subjected to a filtration process, wherein any solid components, including the remnants of the capsules, are effectively removed. What remains is a filtrate containing the extracted VB_12_, now separated from the capsule constituents and ready for further analysis or application [145,146]. The ultrasonication method harnesses high-frequency sound waves to disrupt the capsules and liberate their contents into the surrounding solvent [99]. Ultrasonication proves to be particularly efficient when dealing with nanocapsules, as its energy waves effectively penetrate the smaller dimensions, facilitating the release of VB_12_ encapsulated within these structures [89]. Centrifugation is a valuable technique for extracting VB_12_ from micro and nanocapsules. This method is applied to effectively segregate the solid components of the capsules from the VB_12_-containing solution (un)treated by ultrasonication [94]. Following centrifugation, the resultant mixture undergoes phase separation, with the denser solid components settling at the bottom, leaving the VB_12_-enriched supernatant at the top. It is imperative to carefully collect this supernatant, as it is anticipated to contain the VB_12_ that has been successfully released from the capsules during the process [21]. In the chemical extraction method, the choice of chemicals is depended on the specific composition of the capsules. When dealing with polymer-based capsules, common chemicals used for dissolution include organic solvents (like chloroform, methanol, ethanol, or acetone), dilute acids (such as hydrochloric acid or acetic acid), and basic solutions (such as sodium hydroxide or ammonia) [25,126]. The enzymatic digestion method finds widespread application within the food and pharmaceutical industries due to its ability to efficiently and precisely break down the capsule materials, liberating the encapsulated VB_12_ while maintaining the integrity of the compound of interest [90,147,148]. When dealing with capsules that exhibit a high degree of resistance, supercritical fluid extraction presents an effective solution. This method harnesses supercritical carbon dioxide (CO_2_) or other suitable fluids to exert controlled pressure and temperature, resulting in the breakdown of the capsules and the release of VB_12_ [149]. Supercritical fluid extraction is a favored approach in situations where conventional methods may prove less effective, ensuring efficient capsule disruption and VB_12_ liberation [150]. The selection of an extraction method is contingent upon the specific attributes of the capsules and the designated utilization of the extracted VB_12_. The paramount criterion is the assurance of an extraction approach that adeptly achieves both efficiency in capsule disruption and the preservation of VB_12_’s structural and chemical integrity. Concurrently, it is imperative to adhere to rigorous safety protocols when handling chemicals and equipment during the extraction process, thereby safeguarding the quality and safety of the resultant VB_12_ extract.

## 7. Emerging Biosensing Technologies for Vitamin B_12_ Detection

Analytical methods for detecting VB_12_ in serum and food products typically involve a combination of techniques to ensure precise quantification. In the case of serum analysis, the most common method is immunoassay, particularly the enzyme-linked immunosorbent assay (ELISA). ELISA kits use specific antibodies to detect and measure VB_12_ levels. High-performance liquid chromatography (HPLC) is also widely used, providing precise separation and quantification of VB_12_ in serum [43]. For food products, microbiological assays, such as the *Lactobacillus leichmannii* assay, can be employed to assess VB_12_ content. Additionally, liquid chromatography–mass spectrometry (LC-MS) is increasingly utilized for both serum and food analysis due to its sensitivity and specificity, especially when dealing with complex matrices. These methods, when carefully selected and validated, ensure reliable measurements of VB_12_ levels in various sample types, contributing to reliable nutritional assessments and research outcomes [43,147,151].

Biosensing platforms have been recently applied to accurately detect VB_12_ in various samples. Detecting VB_12_ in various matrices, including serum, supplements, and food products, is essential for both clinical diagnosis and nutritional analysis [147]. Biosensors offer several notable advantages over traditional methods like ELISA and HPLC for detecting VB_12_. They provide rapid results, often within minutes, making them ideal for point-of-care applications and real-time monitoring. Their portability allows for on-site testing, reducing the need for sample transportation to centralized laboratories, which is particularly beneficial in remote or resource-limited settings. Biosensors also achieve high sensitivity and specificity, comparable to ELISA and HPLC, thanks to their use of specific receptors like antibodies or aptamers. Additionally, biosensors can detect multiple analytes simultaneously, require smaller sample volumes, and offer real-time monitoring capabilities [32,152,153]. Therefore, biosensing platforms are versatile tools for VB_12_ assessment in diverse applications, from clinical diagnostics to nutritional research.

In serum, electrochemical biosensors demonstrate exceptional sensitivity. By immobilizing specific receptors on an electrode surface, the binding of VB_12_ to these receptors induces measurable electrochemical changes, enabling rapid and precise serum analysis [32,154]. Additionally, optical biosensors, such as surface plasmon resonance (SPR) and fluorescence-based assays, can detect VB_12_ by monitoring refractive index or fluorescence intensity alterations when the vitamin interacts with recognition elements. These methods provide flexibility and sensitivity for serum VB_12_ detection in clinical settings [154]. Supplements and food products demand different biosensing approaches. Immuno-biosensors, utilizing antibodies or aptamers, are particularly effective in supplement analysis. They rely on the formation of antigen–antibody complexes when VB_12_ is present, facilitating sensitive and specific detection. In food products, colorimetric assays offer a practical choice, producing visible color changes upon VB_12_ interaction with reagents [155,156]. Additionally, nanomaterial-based sensors and microfluidic devices can enhance sensitivity and adaptability in food product testing, particularly when dealing with complex matrices. The versatility of these biosensing platforms ensures the accurate assessment of VB_12_ in diverse samples, enabling improved healthcare and nutritional monitoring [157]. Moreover, chemiluminescence assays can be employed in both clinical and food product analysis. These assays harness the emission of light from chemical reactions, generating detectable signals when VB_12_ is present [158]. Surface-enhanced Raman spectroscopy (SERS), utilizing enhanced Raman signals from nanoparticles, could also offer a unique fingerprint for VB_12_ detection, particularly in complex food matrices [159,160]. In the following sections, an overview of newer and more practical biosensing platforms (such as fluorescent and electrochemical) for the detection of VB_12_ in food products and pharmaceuticals will be provided.

### 7.1. Fluorescent Carbon Dots Nanosensors of Vitamin B_12_

Carbon dots (CDs, <10 nm) have emerged as highly suitable fluorescent bio-probes owing to a range of advantageous properties including excellent biosafety, straightforward preparation and modification processes, intriguing photoluminescence characteristics, and exceptional aqueous dispersibility. CDs find widespread application in biomedicine and sensing technology, often serving as transducing elements or electrode modifiers, either independently or in conjunction with other nanomaterials. Their unique blend of electrical, mechanical, and optical properties contributes to the development of exceptional miniaturized sensors for point-of-care testing [161].

These sensors have garnered significant recent attention for their potential to track VB_12_ levels in both biological and food systems. Kalaiyarasan et al. [162] successfully synthesized highly stable and hydrophilic carbon quantum dots (CQDs) with an impressive fluorescence quantum yield of 43% by involving the dehydration/condensation and aromatization/carbonization of a combination of diethylenetriamine and trisodium citrate. A notable quenching effect was exhibited by VB_12_ on the fluorescence of CQDs. This discovery forms the foundation for a novel quenchometric method, which demonstrates remarkable versatility across a wide range of VB_12_ concentrations, spanning from 1 nM to 20 μM. The discovered approach boasts an impressively low limit of detection (LOD) of 210 pM. These results collectively underscore the potential of the developed method as a sensitive and precise tool for quantifying VB_12_ in various applications [162]. Preethi et al. developed a highly efficient sensor for the detection of VB_12_. This innovative sensor was crafted through a green synthesis approach, employing ultrasonication to create CQDs using curry berry (Murrayakoenigii) extract as the source material. A fascinating dimension of this investigation revolved around the intriguing interaction observed between CQDs and VB_12_. This interaction was primarily attributed to the structural Co^2+^ component within cobalamin. Notably, this interaction led to a gradual reduction in fluorescence intensity, which occurred under specific conditions, spanning a concentration range from 0 to 0.40 μM. Impressively, the method employed in this study achieved a remarkably low LOD of 0.04 μM [163]. These findings shed light on the potential of this sensor for sensitive VB_12_ detection and underline its significance in the field.

Huang et al. developed a straightforward room temperature method to synthesize orange-emitting CDs (O-CDs) via a Schiff base crosslinking reaction between methyl-p-benzoquinone and triethylenetetramine. These O-CDs exhibited strong excitation-dependent emission characteristics with a relative quantum yield of approximately 6.56%. Leveraging the robust inner filter effect, O-CDs were found to be highly effective for the sensitive detection of VB_12_. The intensity ratio (F/F_0_) of O-CDs demonstrated a linear relationship with VB_12_ concentration (50 nM–200 μM) and a low LOD of 10 nM [164]. A one-pot hydrothermal process was applied to synthesize CDs by harnessing their unique properties for an innovative sensing technique known as the inner filter effect (IFE). An interesting alignment emerged—both the maximum absorption peak of VB_12_ and the excitation maxima of CDs converged at 360 nm. This convergence allowed VB_12_ to absorb the emitted light from the excited CDs, resulting in a discernible reduction in the CDs’ fluorescence. As the concentration of VB_12_ increased, a proportional decrease in the fluorescence intensity of the CDs was observed. The developed IFE-based sensing strategy demonstrated a robust linear correlation between the normalized fluorescence intensity and VB_12_ concentration, spanning from 0 to 60 μM. A low LOD of 0.1 μM, attaining a signal-to-noise ratio of 3 was achieved [158]. In a similar technique, Wang et al. synthesized N,P-codoped CDs, based on precursors of _L_-arginine and phosphoric acid, exhibiting excellent water solubility and exceptional luminescent properties as well as boasting a high fluorescence quantum yield of 18.38%. The CDs’ emission intensity peaks at 444 nm when excited at 340 nm. Leveraging the mechanism of the IFE, they harnessed the N,P-codoped CDs as an efficient and selective fluorescence sensor for the detection of VB_12_. Impressively, this method extends the linear response range of CDs to VB_12_, spanning from 2.0 to 98.6 μM and further from 98.6 to 176 μM, with a low LOD of 59 nM. The versatility of the applied approach was demonstrated by successfully detecting VB_12_ in two distinct types of vitamin pills and blood serums. Furthermore, it proved to be a valuable tool for bioimaging HeLa cells, showcasing the broader potential of this sensing platform in biomedical applications [165]. In a study by Fan et al. [166], nitrogen and sulfur co-doped CDs (N,S-CDs) were synthesized through a straightforward hydrothermal method employing o-phenylenediamine and thiourea as precursors. Notably, the primary emission peak at 565 nm was achieved with an excitation wavelength of 420 nm, and the quantum yield reached an impressive 14.3%. A pivotal observation emerged as the absorption spectra of VB_12_ exhibited substantial overlap with the emission peak of the N,S-CDs. Leveraging the principles of the IFE between VB_12_ and N,S-CDs, Fan et al. established a label-free and highly sensitive method for detecting VB_12_ in pharmaceuticals. Under meticulously optimized conditions, the concentration of VB_12_ showcased a robust linear relationship with the fluorescence quenching degree (ΔF/F_0_) of the reaction system at 565 nm. This method effectively spanned a range from 0.25 to 20 μM, offering a very low LOD of 77.5 nM. These N,S-CDs, characterized by their distinctive orange fluorescence emission properties, hold significant promise for diverse applications within the realm of biological analysis [166].

### 7.2. Nanoclusters as Fluorescent Probes of Vitamin B_12_

Precisely engineered noble metal nanoclusters at the atomic level have garnered significant attention for their versatile applications, including sensing and therapeutic aims. Unlike larger plasmonic nanoparticles, these nanoclusters exhibit distinctive molecule-like fluorescence characteristics attributed to their discrete energy levels. Samari et al. devised a simple yet highly sensitive method to swiftly detect VB_12_ by relying on the fluorescence quenching of bovine serum albumin-stabilized gold nanoclusters (BSA-AuNCs). Optimizing studies showed that this method effectively quantifies VB_12_ within the range of 160.0 ng mL^−1^ to 38.5 μg mL^−1^, exhibiting a strong correlation coefficient of 0.998 and a low LOD of 100.0 ng mL^−1^. Also, they successfully applied this approach to analyze commercial injection dosage forms, achieving recovery rates between 97.7 and 102% and relative standard deviations ranging from 2.0 to 5.9%. This method offers a dependable and efficient means for VB_12_ determination [167]. Another research group synthesized water-soluble copper nanoclusters protected by polyethyleneimine (CuNCs@PEI) using a single-step method involving ultraviolet radiation and microwave heating. These CuNCs@PEI served as a robust fluorescent probe for the sensitive detection of tetracycline hydrochloride (TCH) and VB_12_. The developed probe exhibited a linear response within the concentration range of 0.33–66.67 μmol L^−1^ for TCH and 0.33–53.33 μmol L^−1^ for VB_12_, with corresponding LOD calculated at 55.50/56.34 nmol L^−1^. The fluorescence quenching of CuNCs@PEI by TCH/VB_12_ was attributed to a combination of Förster resonance energy transfer (FRET) and the IFE. Additionally, our probe was successfully applied to assess TCH in veterinary powder or TCH tablets, as well as VB_12_ in oral liquid or VB_12_ tablets, yielding satisfactory results when compared to the standard HPLC method. The CuNCs@PEI probe also demonstrated reliable temperature-sensing capabilities [168]. Shanmugaraj et al. successfully prepared histidine-stabilized copper nanoclusters (His-CuNCs) through a straightforward and eco-friendly method, avoiding the use of toxic organic solvents. Then, these His-CuNCs were effectively utilized for the sensitive and selective detection of VB_12_ so that the emission intensity of His-CuNCs exhibited a significant decrease by adding VB_12_. This quenching mechanism was attributed to FRET between His-CuNCs and VB_12_, resulting in an LOD estimated at 3.30 × 10^−9^ mol dm^−3^. Moreover, this approach allowed for the selective determination of VB_12_ even in the presence of other interfering vitamins [169]. Also, the fluorescence emitted by silver nanoclusters, which were templated using hyperbranched polyethyleneimine with varying molecular weights and terminal groups, could be effectively quenched in the presence of VB_12_ under the IFE mechanism [170]. Sarkar et al. have recently shown a simple method for creating red-emitting silver nanoclusters within lysozyme (LYS–AgNCs) scaffolds using dithiothreitol reduction. LYS–AgNCs exhibited uniform size, excellent water solubility, and impressive photoluminescence, making them suitable for dual-mode sensing of Cu^2+^ and VB_12_. Two distinct fluorescence-quenching mechanisms are at play: Cu^2+^-induced quenching involves static and dynamic quenching, while VB_12_-induced quenching relies on the IFE and FRET. LYS–AgNCs were proven effective for real-sample VB_12_ analysis, ensuring their practicality in various applications [171].

### 7.3. Electrochemical Sensors of Vitamin B_12_

Table 3 provides an overview of the latest advancements in electrochemical sensors designed for the detection of VB_12_, including details on electrode composition and modification, sensing methodology, and sensitivity [172,173,174,175,176,177,178,179,180,181,182,183]. Also, the LOD and linear range of VB_12_ recognized by some electrochemical sensors are given in Figure 8. Guo and Yang developed electrochemical sensors using an Au-PPyNPs@f-CNTs nanocomposite to modify a glassy carbon electrode (GCE) for sensitive VB_12_ detection (Table 3). Characterization revealed the nanocomposite’s structure, featuring Au-NPs in an fcc crystal structure and amorphous PPy-NPs on a highly interconnected network of functionalized CNTs. Electrochemical tests demonstrated the electrode’s stable and effective electrocatalytic response to VB_12_, owing to the excellent conductivity of Au, PPy, CNTs, and well-distributed electrochemically active sites (Table 3). The sensor exhibited a linear range of 0 to 85 μM, a low LOD of 0.9 nM, and high sensitivity (4.3597 μA/μM). They validated the sensor’s utility with real pharmaceutical VB_12_ capsules and human plasma samples, leading to accurate results with acceptable recovery and reproducibility values [174].

**Table 3 molecules-28-07469-t003:** Electrode materials and modifications for VB_12_ detection across diverse samples through electrochemical sensing.

Electrode Material	Chemical Modification of Electrode (with)	Electrochemical Technique	Sample Media	Sensitivity	Ref.
Platinum (Pt)	Cu(1,3,5-benzenetricarboxylic acid)(4,4′-bipyridine)·3 (N,N′-dimethylformamide)	Cyclic voltammetry (CV)	Two commercial pharmaceutical VB_12_-tablets	0.104 µA µM^−1^	[172]
Indium tin oxide (ITO)	Gold-tin dioxide (AuSnO_2_)	Differential pulse voltammetry (DPV)	Fresh cow’s milk	-	[173]
Glassy carbon	Gold-polypyrrole nanoparticles and functionalized carbon nanotubes (Au-PPyNPs@f-CNTs)	Amperometry	VB_12_ capsule, Human plasma	4.3597 µA µM^−1^	[174]
Carbon fiber paper	Palladium-gold polypyrrole (PdAu-PPy)	Differential pulse voltammetry (DPV)	Human Blood serum, Urine	10.576 μA μM^−1^ cm^−2^	[175]
Graphenic carbon (GUITAR)	Copper oxide	Linear sweep voltammetry (LSV)	Four biological Samples	-	[176]
Boron-doped diamond	-	Squarewave voltammetry (SWV)	VB_12_ supplementation tablets, VB_12_-fortified toothpaste	4.17 μA μM^−1^ cm^−2^	[177]
Gold	Polypyrrole/ferromagnetic nanoparticles/triazine dendrimer (PPy/FMNPs@TD)	Differential pulse voltammetry (DPV)	Food product	25.6 (μA/μM)	[178]
Glassy carbon	Poly(thionine) film	Cyclic voltammetry (CV)	Injection sample	-	[179]
Pencil graphite	Poly(3,4-ethylenedioxythiophene/silver nanoparticles (PEDOT)/AgNPs	Differential pulse voltammetry (DPV)	Human blood serum, Urine	-	[180]
Glassy carbon	Polymethylene blue/zinc oxide nanoparticles (polyMB/ZnONPs)	Differential pulse voltammetry (DPV)	Two commercial pharmaceutical VB_12_-tablets	-	[181]
Carbon paste	[Mn(thiophen-2-carboxylic acid)2(triethanolamine)]	Squarewave voltammetry (SWV)	Commercial VB_12_-tablets, Dietary VB_12_-supplements	-	[183]
Pencil graphite	Methyl blue (MB)-adsorbed reduced graphene oxide (rGO) and functionalized multiwalled carbon nanotubes (f-MWCNTs)/acryloylurea-molecularly imprinted polymer (AU-MIP)	Differential pulse anodic voltammetry (DPAV)	Pharmaceutics, Blood serum, Urine, Cerebrospinal fluid (CSF)	-	[182]

Manivel et al. developed an ultrasensitive electrochemical sensor for VB_12_ detection using a Pt-modified Cu(HBTC)(4,4′-bipy)·3DMF electrode as a highly efficient electrocatalyst. The Cu complex nanorods were synthesized via solvothermal methods. Cyclic voltammetric (CV) analysis revealed the modified electrode’s excellent electrocatalytic redox reversibility toward the Co^3+/^Co^2+^ redox couple, offering a high sensitivity of 0.104 µA µM^−1^, a low 50 nM LOD, and a wide linear range of 0.1–188.2 µM. This sensor’s novelty lies in using both oxidation and reduction currents of the Co^3+/^Co^2+^ redox couple, reducing analytical errors. The enhanced electrocatalytic capability allowed for discrimination of VB_12_ from potential interfering species, making it suitable for analyzing VB_12_ in pharmaceutical tablets [172]. In another study, Kou et al. fabricated a poly(thionine) (PTH) film on an electrochemically activated GCE using a two-step CV scan. Hydroxyl radicals, generated in a Fenton-like reaction, effectively oxidized PTH under near-neutral conditions, enhancing the cathodic peak current. The addition of VB_12_ to the Cu^2+^–H_2_O_2_ system inhibited PTH oxidation due to copper ions binding with VB_12_ ligands and the reduced catalytic ability of Co^2+^ for hydroxyl radical generation. The cathodic peak current change was linear with the logarithm of VB_12_ concentration ranging from 10 nmol L^−1^ to 100 μmol L^−1^, with a 2 nmol L^−1^ detection limit under optimal conditions. The developed sensor demonstrated high sensitivity, selectivity, reproducibility, and stability. VB_12_ content in the injection sample was measured, yielding recovery data between 92 and 102% [179].

Figure 9 signifies that cyclic voltammograms depict thionine polymerization on GCE(ea) with gradually increasing anodic and cathodic peak currents, signifying PTH film formation, characterized by two redox couples in the 30th CV curve. The SEM image reveals a homogeneous film resembling a neural network structure, confirming the presence of PTH on the electrode surface. The FTIR analysis also shows absorption bands at ∼1386 and ∼1655 cm^−1^ attributed to C=C and C=N stretching vibrations, and an absorption band at ∼3387 cm^−1^ indicating N–H stretching vibrations of −NH_2_, confirming the presence of thionine units in the PTH film. Moreover, CV curves at different scan rates demonstrate rising redox peak currents and shifts in anodic and cathodic peak potentials, indicating a surface-controlled electrode process on the GCE(ea)/PTH surface (Figure 9).

Table 3 shows that most researchers utilized the differential pulse voltammetry (DPV) technique using various electrodes and their modification methods to analyze VB_12_ [173,175,178,180,181]. Moreover, Singh et al. introduced a novel three-dimensional biomimetic imprinted polymer tailored for the selective detection of VB_12_ using the technique of differential pulse anodic voltammetry (DPAV). They employed a composite comprising methyl blue-adsorbed reduced graphene oxide (rGO) and functionalized multiwalled carbon nanotubes (f-MWCNTs) with acryloylurea functionalization. This composite boasts superior electroconductivity and an expanded surface area when compared to pristine carbon nanotubes or graphene. Notably, this marks a pioneering endeavor, as it represents the first synthetic biomimetic polymer designed specifically for VB_12_ grown onto the surface of a pencil graphite electrode. The integration of methyl blue with rGO sheets not only enhances solubility but also improves conductivity and self-assembly properties. This unique combination of rGO sheets and f-MWCNTs significantly enhances kinetics and conductivity within the molecularly imprinted polymer structure. To achieve ultratrace detection of VB_12_, researchers employed differential-pulse voltammetric transduction. The developed sensor demonstrates exceptional sensitivity and selectivity for VB_12_, particularly in real sample scenarios [182]. Pereira et al. [177] and Karastogianni et al. [183] both applied square wave voltammetry (SWV) as a detection method for VB_12_. Pereira et al. utilized boron-doped diamond electrodes, while Karastogianni et al. employed a carbon paste electrode with a manganese complex film modified with thiophene-2-carboxylic acid and triethanolamine as ligands (Table 3). In contrast, Tian et al. utilized the electrochemical process involved using a plain graphene electrode to convert VB_12_ molecules’ nitrogen groups to NO^3−^ at 1.3 V vs. Ag/AgCl for 15 min. VB_12_ quantification relied on measuring the resulting oxidized nitrate anions, which were then reduced using a copper oxide nanocrystal-decorated graphene electrode. Cathodic polarization with a graphite rod electrode was conducted before nitrate reduction to eliminate potential interferences. Under optimized conditions, this approach provided a wide linear detection range of 0.15–7378 nmol L^−1^ with an LOD of 0.59 nmol L^−1^ (Table 3). Interestingly, results from biological samples closely matched those obtained via the HPLC method [176].

## 8. Conclusions and Future Remarks

This study has provided an overview of recent advancements in the realm of VB_12_, encompassing dietary sources, health advantages, encapsulation methodologies, strategies for food fortification, and biosensing platforms. Findings showed that VB_12_ is primarily abundant in animal-derived sources. While trace amounts of analogs of this vitamin are found in certain plant-based sources, they are not reliable for meeting human dietary requirements. Therefore, individuals following strict vegetarian or vegan diets are recommended to obtain VB_12_ from fortified foods or supplements. Developing new encapsulation and fortification strategies are needed to increase the absorption and bioavailability rate of VB_12_, prevent megaloblastic anemia, and support the healthy functioning of the nervous system. Microencapsulation techniques of VB_12_ enhance the molecule’s physicochemical stability, bioavailability, and controlled delivery. Nanoencapsulation in terms of NVs, NEs, and NPs using highly efficient biopolymers and hydrocolloids along with VB_12_ surface functionalization can significantly improve its intestinal transport for achieving maximum bio-functionality. Innovative approaches to food fortification, encompassing the enrichment of salt, sugar cubes, and wheat-based products with both free and encapsulated VB_12_, have shown promising effectiveness. Nevertheless, there has been a growing emphasis on fortifying dairy and bakery products with encapsulated VB_12_ alone or other micronutrients that promise to combat dietary deficiencies and enhance overall nutritional intake. Also, fluorescent (mainly, nanocluster-based fluorescent probes and fluorescent CD nanosensors) and electrochemical biosensing platforms could be efficiently utilized to determine VB_12_ with high sensitivity and specificity. These biosensors proved results consistent with VB_12_ levels obtained through HPLC analysis in real samples.

Given the successful industrial production of VB_12_ using specific bacteria in fermentation bioreactors, there is a need to explore new microbial species capable of synthesizing different cobalamin types. Additionally, a comprehensive evaluation of the correlation between serum VB_12_ levels, other hematological indices, serum iron levels, and methylmalonic acid levels across all population groups, particularly those with low VB_12_ intake, is warranted. Metabolomic studies, including investigating the role of gut microbiota composition in individuals with both sufficient and deficient serum VB_12_ levels holds the potential to shed light on the gut–brain axis’ involvement in the development of chronic and underlying diseases.

Future investigations into micro- and nanoencapsulation of VB_12_ should prioritize advancements in encapsulation techniques to maximize both yield and EE. The exploration of innovative encapsulation materials and methodologies, including biodegradable polymers and lipid-based carriers, holds the potential to significantly bolster stability and bioavailability. To validate the practical effectiveness of encapsulated VB_12_ in fulfilling dietary needs, it is imperative to extend research from in vitro bioavailability assessments to comprehensive in vivo evaluations. Additionally, delving into the intricate release kinetics of VB_12_ from encapsulated structures during the digestive process promises to yield valuable insights for future applications. Future research in the production of VB_12_-enriched bread and bakery products should focus on optimizing fermentation conditions, including variables like time, temperature, and lactic acid bacteria (LAB) strains, to maximize VB_12_ yields. Additionally, assessing the sensory attributes and consumer acceptance of these fortified products is crucial to ensure their market success. Furthermore, investigating the interactions between VB_12_-producing LAB strains in sourdough and gut microbiota can shed light on their impact on gut health. Long-term studies are needed to evaluate the health benefits of regular consumption of such products in preventing deficiency-related issues. Finally, exploring alternative fortification techniques, such as encapsulation, can enhance VB_12_ stability and bioavailability in bakery items. These research directions will contribute to the development of effective strategies for combating VB_12_ deficiencies through bakery products.

Innovations in detecting VB_12_ using biosensors should prioritize several key areas for future research. Firstly, there is a pressing need to enhance the sensitivity of biosensors to enable accurate detection of low concentrations of VB_12_, particularly in clinical and food samples. Additionally, exploring multiplex detection capabilities to simultaneously assess multiple vitamins and nutrients would provide a comprehensive nutritional profile. Real-time monitoring biosensors that continuously track VB_12_ levels could offer dynamic insights into absorption and metabolism. Portable, user-friendly point-of-care biosensor devices should be developed for rapid and convenient testing in clinical and field settings. Integration with the Internet of Things (IoT) for remote monitoring and data sharing could enhance accessibility to VB_12_ status information. Ensuring biocompatibility for in vivo applications, improving long-term stability, and validating biosensors across diverse sample types are also crucial. Furthermore, affordability should be a focus to ensure widespread adoption, and interdisciplinary collaborations can drive innovation in this field, ultimately benefiting healthcare and research.

## Figures and Tables

**Figure 1 molecules-28-07469-f001:**
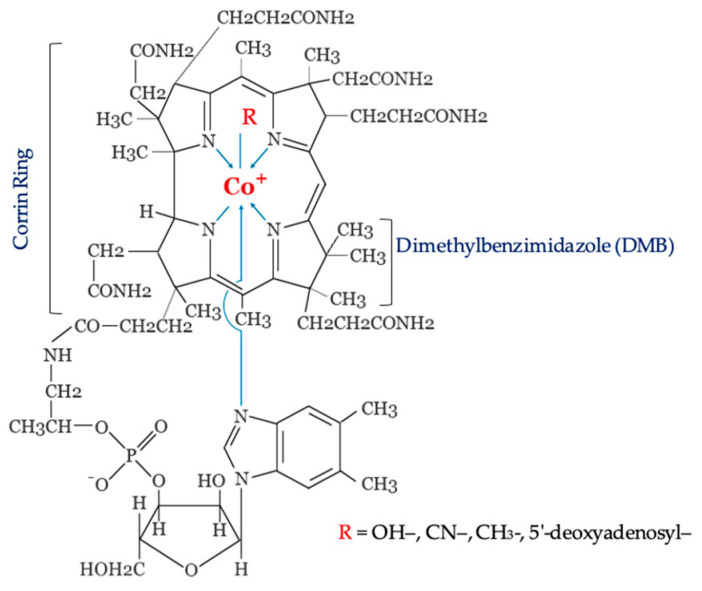
Chemical structure of VB_12._ (Co^+^ is central cobalt ion linked to the upper ligand (R).

**Figure 2 molecules-28-07469-f002:**
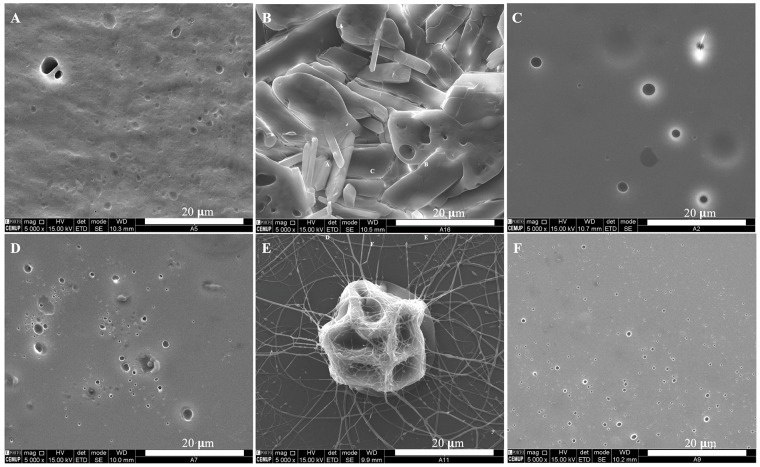
Scanning electron microscopy (SEM; magnification of 5000×, scale bar of 20 μm) images of electrospun and electrosprayed VB_12_-loaded zein microstructures (70% ethanol, 10–20% zein (Z), and 1–10% VB_12_) prepared by electrospinning and spray-drying techniques under different operating and formulation conditions: (**A**), 10 Z:1 VB_12_; (**B**), 10 Z:5 VB_12_; (**C**), 10 Z:10 VB_12_; 0.3 mL/h of flow rate, 7 cm distance; 20 Z:5 VB_12_; 0.2 mL/h of flow rate with 7 cm (**D**), 10 cm (**E**), and 15 cm (**F**) distances). Reprinted with permission from [90].

**Figure 3 molecules-28-07469-f003:**
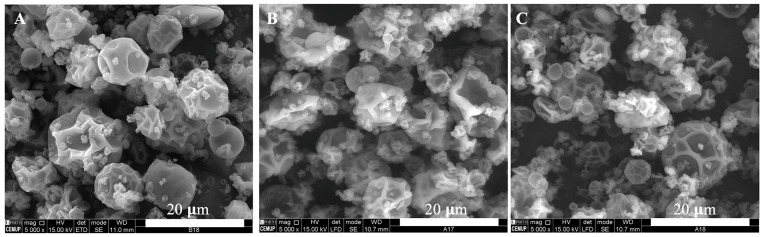
SEM images (magnification of 5000×, scale bar of 20 μm) of 20% zein (Z)-based microcapsules loaded with VB_12_ (1–10%) and prepared by spray-drying: (**A**), Z20:1 VB_12_; (**B**), Z20:5 VB_12_; (**C**), Z20:10 VB_12_). Reprinted with permission from [90].

**Figure 4 molecules-28-07469-f004:**
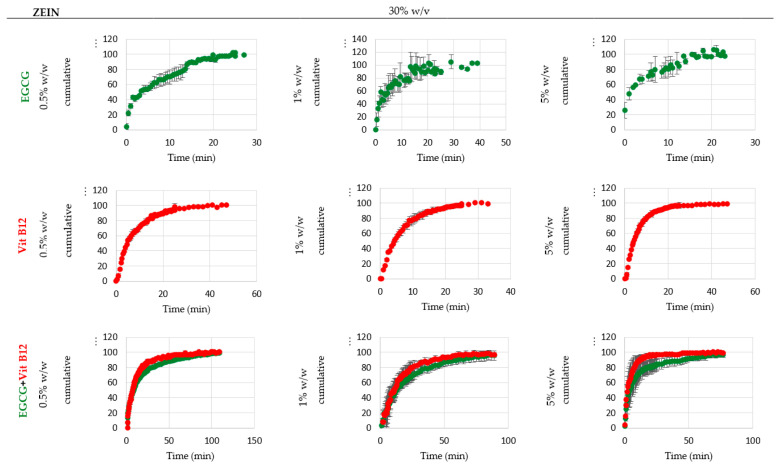
In vitro release profiles of EGCG/VB_12_/(EGCG + VB_12_) normalized by the total amount released, in water of the electrospun zein (30% *w*/*v*) microstructures loaded with active compounds (0.5, 1, and 5% *w*/*w*). Reprinted from [83].

**Figure 5 molecules-28-07469-f005:**
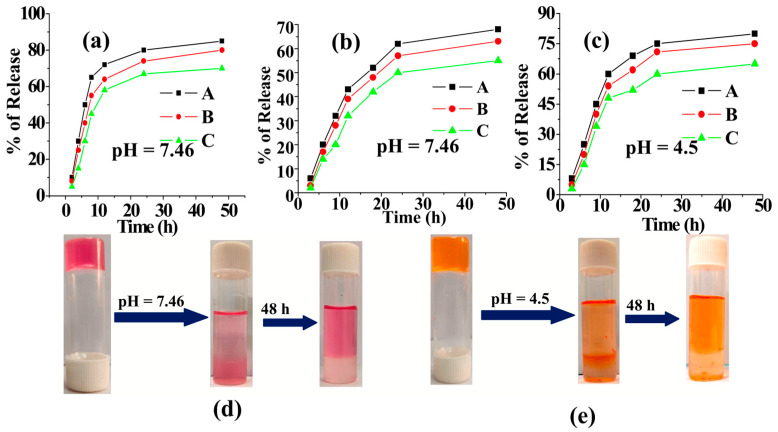
The release profile of VB_12_ at pH 7.46 (**a**), doxorubicin at pH 7.46 (**b**), and doxorubicin at pH 4.5 (**c**). The pictorial representation of release of VB_12_ (**d**) and doxorubicin (**e**). Reprinted with permission from [102].

**Figure 6 molecules-28-07469-f006:**
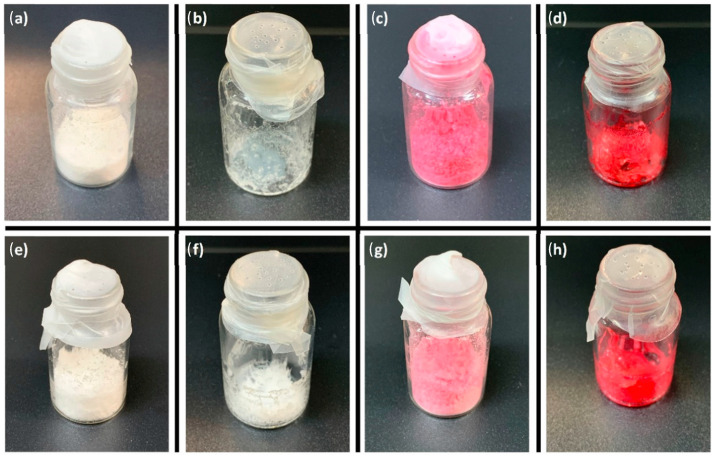
Representative images of lyophilized VB_12_ lipid vesicles: (**a**) empty liposome with lactose, (**b**) empty liposome with sorbitol, (**c**) VB_12_-loaded liposome with lactose, (**d**) VB_12_-loaded liposome with sorbitol, (**e**) empty transfersome with lactose, (**f**) empty transfersome with sorbitol, (**g**) VB_12_-loaded transfersome with lactose, and (**h**) VB_12_-loaded transfersome with sorbitol. Reprinted from [107].

**Figure 8 molecules-28-07469-f008:**
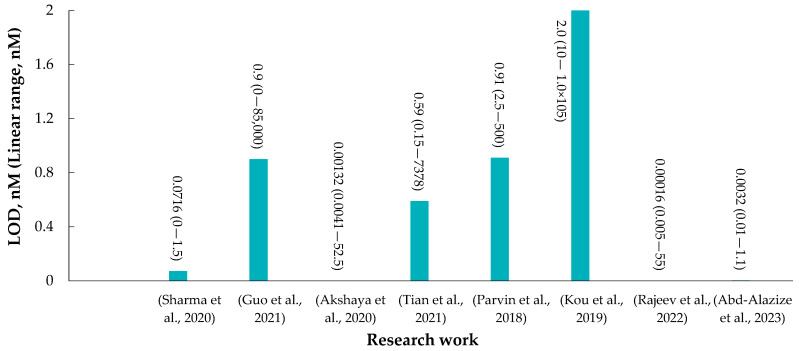
The limit of detection (LOD) and linear range of VB_12_ detected by some electrochemical sensors (see Table 3 for more information) [173,174,175,176,178,179,180,181].

**Figure 9 molecules-28-07469-f009:**
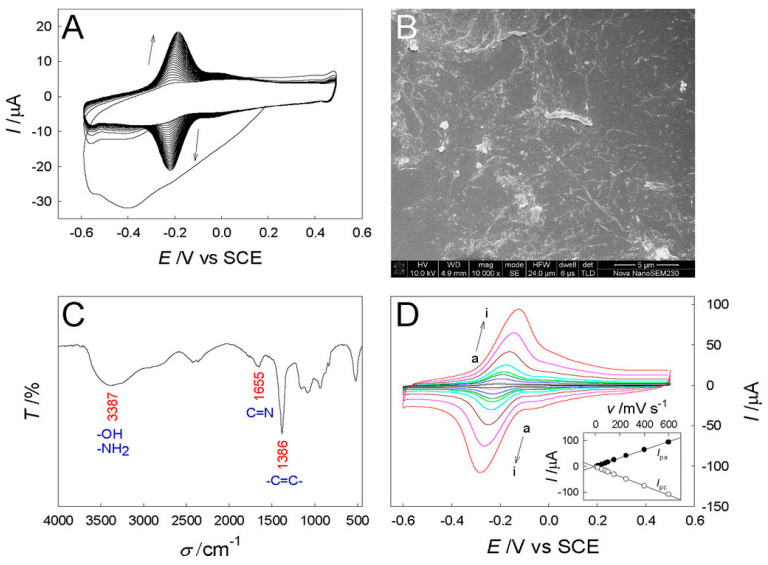
(**A**) Cyclic voltammograms recorded during the growth process of the PTH film on GCE in a nitrogen-saturated 0.1 M pH 6.0 buffer solution containing 5 mmol L^−1^ TH and 0.1 mol L^−1^ NaNO_3_, with a scan rate of 100 mV s^−1^. (**B**) SEM image of the GCE(ea)/PTH surface. (**C**) FTIR spectrum of the PTH film (The wavenumbers corresponding to the primary chemo-functional groups (−OH, −NH_2_, C=N, and −C=C−) are highlighted in red). (**D**) Cyclic voltammograms of GCE(ea)/PTH in 0.1 mol L^−1^ pH 6.5 buffer at various scan rates (ν, color curves), ranging from a to i: from 10 to 600 mV s^−1^. Reprinted with permission from [179].

**Table 1 molecules-28-07469-t001:** A list of the main food sources abundant in fat-soluble vitamins along with their RDI values and the body’s physiological functions.

Fat-Soluble Vitamin Type	Systematic Name	Food Source(s)	RDI (for Adults *)	Function(s)	Ref.
A	Retinol	Animal sources: liver, fish liver oils, eggs, and dairy products. Plant sources: beta-carotene (as precursor of vitamin A in orange, yellow, and green leafy vegetables, such as carrots, sweet potatoes, and spinach)	750 μg (men), 650 μg (women)	Vision improvement, Corneal and conjunctiva development, Immune system functioning, Skin health, Cellular growth and differentiation, Bone and fetus development, Central nervous system formation	[1]
D	Calciferol	Natural sources: fatty fish (salmon, mackerel, tuna), fish liver oils, and egg yolks. Fortified sources: fortified dairy products, orange juice, and cereals.	15 μg (both genders)	Calcium absorption, Bone health and skeletal muscle function, Immune function, Antiviral activity, Reducing cytokine release and adipose tissue inflammation	[2,3]
E	Tocopherol	Nuts and seeds: walnuts, almonds, sunflower seeds, and hazelnuts. Plant oils: wheat germ oil, palm and rice bran oils, sunflower oil, and safflower oil. Green vegetables: Spinach and broccoli.	15 mg (both genders)	Antioxidant and anti-inflammatory effects, Immunity, Skin health, potent antioxidant characteristics, Platelet coagulation inhibition, Cellular signaling, Lowering cholesterol	[4,5]
K	Phylloquinone	Leafy greens: kale, spinach, collard greens, and Swiss chard. Vegetables: Brussels sprouts, broccoli, and asparagus. Plant oils: olive, canola, and soybean. Fish: certain types of fish, such as mackerel and salmon.	120 μg (men), 75–90 μg (women)	Blood clotting, Bone health, Cardiovascular health benefits	[6]

* Adults: Men and nonpregnant women.

**Table 2 molecules-28-07469-t002:** A summary of the major food sources rich in some water-soluble vitamins accompanied with their RDI values and the body’s physiological functions.

Water-Soluble Vitamin Type	Systematic Name	Food Source(s)	RDI (for Adults *)	Function(s)	Ref.
C	Ascorbic acid	Citrus fruits, Berries, Kiwi, Tropical fruits, Guava, Star and jujube fruits, Black currant, Strawberry, Melons, Paprika, Tomatoes, Leafy greens, Broccoli, Potatoes, and Cauliflower.	90 mg (men), 75 mg (women)	Antioxidant activity, Collagen formation, Immune system support, Iron absorption, Neurotransmitter production, Wound healing, Skin health, Cardiovascular health	[7,8]
B_1_	Thiamine	Whole grains, Legumes, nuts, Seeds, Pork, and Fortified cereals	1.2 mg (men), 1.1 mg (women)	Energy metabolism, Nerve cells functioning, Muscle contraction	[9,10,11]
B_2_	Riboflavin	Dairy products, Lean meats, Green leafy vegetables, Yeast extract, Almonds, Enriched cereals, Eggs, and Mushrooms.	1.3 mg (men), 1.1 mg (women)	Energy production via the electron transport chain, Antioxidant activity (in cellular respiration and the immune system), Maintaining healthy skin, eyes, and nerve functions	[9,11,12]
B_3_	Niacin	Poultry, Fish, Peanuts, Mushrooms, and Fortified Cereals	16 mg (men), 14 mg (women)	Energy metabolism, Synthesis of fatty acids, Maintaining healthy skin, nerves, and digestive system	[9,11]
B_5_	Pantothenic acid	Meat, Whole Grains, Legumes, and Vegetables	5 mg (both genders)	A component of coenzyme A, involving in various metabolic processes, such as the synthesis of fatty acids and the citric acid cycle (Krebs cycle)	[9,11]
B_6_	Pyridoxine	Chicken, Fish, Bananas, Potatoes, and Fortified Cereals	1.0–1.7 mg (both genders)	Amino acid metabolism, Synthesis of neurotransmitters (e.g., serotonin and norepinephrine), Immune function, Formation of red blood cells	[9,13]
B_7_	Biotin	Egg Yolks, Nuts, Seeds, and Some Vegetables	30 μg (both genders)	Involved in various metabolic reactions, including the breakdown of fatty acids and amino acids, Essential for healthy skin, hair, and nails	[9,14]
B_9_	Folate/folic acid	Leafy greens (e.g., spinach), Legumes, Citrus fruits, and Fortified grains	400 μg (both genders)	DNA synthesis, Cell division, Formation of red blood cells, Preventing neural tube defects in the developing fetus	[9,13,14]
B_12_	Cobalamin	Animal-based Foods (such as Meat, Dairy Products, and Eggs)	2.4 μg (both genders)	DNA synthesis, Formation of red blood cells, Nerve function, Maintaining the health of the nervous system	[9,13]

* Adults: Men and nonpregnant women.

## Data Availability

Not applicable.

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
