# Peer review of "Recent Advances in Dietary Sources, Health Benefits, Emerging Encapsulation Methods, Food Fortification, and New Sensor-Based Monitoring of Vitamin B12: A Critical Review"

_molecules, 2023, doi:10.3390/molecules28227469_

Round 1

Reviewer 1 Report

Comments and Suggestions for Authors

The MS entitled “Recent Advances in Dietary Sources, Health Benefits, Emerging Encapsulation Methods, Food Fortification, and Newly Sensor Based Monitoring of Vitamin B12: A Critical Review” by Seyed Mohammad K. et al. has been reviewed. This topic is important, and the authors provide a critical review relating to dietary sources, health benefits, emerging encapsulation methods, food fortification, and newly sensor based monitoring of vitamin B12. The following comments/suggestions are given for specific improvements that should be made prior to publication.

1.      Please delete or find other phrases for “e.g.” as it is repeated so many times in the MS.

2.      Check the number of words allowed in the abstract.

3.      Letters in a sentence should be the same size (lines 124, 181, 182, 246, 356, ..)

4.      Line 125: “Fig.1” change to “Figure 1”

5.      Attention should be paid to SI units. (For example microgram: mcg should be change to μg).

6.      I would suggest that the authors make a table in which they will clearly show the fat- and water-soluble vitamins and the amounts and the foods in which they are found.

7.      I would suggest creating a scheme such as an encapsulation scheme.

8.      Table 1 should be divide into more tables (Table 2,..), it is very confusing.

9.      Please rewrite keywords (for example diagnostic biomarkers?).

10.   The resolution of Figure 1 should be improved.  Abbreviation DMB? –add full name. I would suggest marking R, Co, and DMB with different colors.

11.   I would suggest that authors add an abbreviation (VB12 ) for vitamin B12 when it is the first time mentioned in the text.

Author Response

Please find the response to Reviewer 1 in the attachment. 

Reviewer 2 Report

Comments and Suggestions for Authors

This is a comprehensive review on VB12. The manuscript is well-written and I only have two concerns.

1. Introduction: The authors should stress the importance/advantage of encapsulation of VB12 in a different perspective. There are existing products that has been widely used for nutitional fortification, such as vitamin pills.

2. When encapsulating VB12, how is the ingredient extracted? I believe it could enrich the content of this manuscript if the authors could also report the extraction techniques of VB12.

Author Response

Please find the response to Reviewer 2 in the attachment. 

Reviewer 3 Report

Comments and Suggestions for Authors

The manuscript is interesting and well-prepared. It provides an overview of recent reports concerning vitamin B12 - dietary sources, biological activity, methods of encapsulation, and food fortification with this vitamin. I recommend the article for publication.

In my opinion, the topic is original. It addresses a specific gap in the field of public health. The references were appropriate. The novelty of the manuscript is significant - there is no other manuscript that addresses the issue. The figures are of good quality.

Author Response

Please find the response to Reviewer 3 in the attachment. 

Round 2

Reviewer 1 Report

Comments and Suggestions for Authors

Dear Authors,

All of my comments have been thoroughly addressed in the revised version of the MS. I appreciate the revisions made and recommend accepting the MS in its current form.